# Rare jackpot individuals drive rapid adaptation in Threespine Stickleback

Alexander Kwakye [1,2], Kerry Reid [3], Matthew A. Wund [4], David C. Heins [5], Michael A. Bell[6] & Krishna R. Veeramah [1] ✉

Recombination has long been considered the primary mechanism to bring beneficial alleles together, which can increase the speed of adaptation from standing genetic variation. Recombination is fundamental to the transporter hypothesis proposed to explain precise parallel adaptation in Threespine Stickleback. We study an instance of freshwater adaptation in the Threespine Stickleback system using whole genome data from an evolutionary time-series to observe the genomic dynamics underlying rapid parallel adaptation. Here, we show that rapid adaptation to a freshwater environment depends on a few individuals with large haploblocks of freshwater-adaptive alleles (jackpot carriers) present among the anadromous founders at low frequencies. Biological kinship analyses indicates that mating among jackpot carriers and between jackpot carriers and non-jackpot individuals led to an increase in freshwater-adaptive alleles within the first few generations. This process allowed the population to overcome a substantial bottleneck likely caused by the low fitness of first-generation stickleback possessing a few freshwater-adaptive alleles born in the lake. Additionally, we find evidence that the genetic load that emerged from population growth after the bottleneck may have been reduced through an increase in homozygosity by inbreeding, ultimately purging deleterious alleles. Recombination likely played a limited role in this case of very rapid adaptation.

Darwin originally proposed that evolution by natural selection is a gradual process whereby changes in phenotypes that led to speciation occur by selection of variants with small fitness effects over millions of years[1]. However, there are now numerous examples of rapid evolution, i.e., within tens of generations[2–5], which provide an opportunity to study the molecular basis of evolution in more complex eukaryotes outside of a lab setting.

Rapid adaptation proceeds predominantly from standing genetic variation (SGV)[6–10], and *Gasterosteus aculeatus* (i.e., Threespine Stickleback fish) is a premier example of this phenomenon[11]. In this species, hundreds of loci, most estimated to be more than a million years old[12],

are the basis for much more recent, precisely repeated, rapid adaptation of ancestral marine or anadromous (i.e., collectively 'oceanic') ecotypes to freshwater habitats[12–14]. This process can occur within a decade[12,15].

According to the transporter hypothesis[16] (see also ref. 10), ancient SGV is maintained at low frequencies in oceanic populations through persistent gene flow from multiple freshwater populations, which are sympatric with anadromous populations during the breeding season. Oceanic-freshwater hybrids with haploblocks (large chromosomal segments with multiple contiguous loci) of freshwater-adaptive alleles move back into the oceanic populations, where they

[1]Department of Ecology and Evolution, Stony Brook University, Stony Brook, NY, USA. [2]The Graduate Program in Genetics, Stony Brook University, Stony Brook, NY, USA. [3]Department of Ecology and Evolutionary Biology, Yale University, New Haven, CT, USA. [4]Department of Biology, The College of New Jersey, Ewing, NJ, USA. [5]Department of Ecology & Evolutionary Biology, Tulane University, New Orleans, LA, USA. [6]University of California Museum of Paleontology, University of California, Berkeley, CA, USA. ✉e-mail: krishna.veeramah@stonybrook.edu

should be disfavored but survive well enough to backcross to anadromous stickleback. Successive generations of backcrossing with oceanic stickleback and recombination break these blocks of freshwater-adaptive alleles apart, and most oceanic individuals are observed to possess only a small number of them, mostly in the heterozygous state[12]. Consequently, adaptation to new freshwater habitats is thought to proceed from colonizing oceanic individuals, each possessing only a few freshwater-adaptive alleles. In freshwater habitats, selection then favors the freshwater-adaptive alleles, leading to an increase in their frequencies, and genetic linkage among these alleles will allow for their re-assembly back into large haploblocks over multiple generations.

Alternatively, rapid freshwater adaptation may involve selection that strongly favors jackpot carriers. Jackpot carriers are rare oceanic individuals that possess large haploblocks of freshwater-adaptive alleles. Previous studies suggest that the frequency of jackpot carriers in marine environments is about 0.1%[12,17]. Bassham et al.[17] proposed that rapid freshwater adaptation may depend on such jackpot carriers. In addition, simulations have found that a few individuals with large haploblocks of freshwater-adaptive alleles within founding oceanic populations would be sufficient to adapt to new freshwater habitats[18]. The presence of such individuals could dramatically increase the speed with which haplotypes of adaptive alleles spread through the population compared to the time required for recombination to bring these adaptive alleles together within individuals. In addition, previous work has shown that recombination is significantly suppressed in regions of the genome containing freshwater-adaptive alleles, allowing the maintenance of what has been termed "adaptive cassettes"[12,19,20].

In this study, we examine a recently founded freshwater population in Scout Lake, Alaska, derived from anadromous ancestors[21]. We sequence hundreds of whole genomes from samples collected during the first few generations of rapid adaptation to conditions in Scout Lake. Our results show that adaptation to the freshwater environment depends on the presence of a few jackpot carriers among the founders. After a bottleneck in the third year after founding, subsequent population growth was driven primarily by breeding among the descendants of those few jackpot carriers, leaving signals of marked inbreeding. The increase in homozygosity from inbreeding facilitated the removal of deleterious alleles, thereby likely improving the population's adaptability. Our empirical study shows that only a few jackpot carriers can drive rapid adaptation of anadromous Threespine Stickleback to new freshwater habitats. These findings also suggest a limited role of recombination during the earliest stages of rapid adaptation, in contrast with predictions from the transporter hypothesis.

## Results

### The Scout Lake experiment

Scout Lake (60.5353 N, 150.8322 W) is on the Kenai Peninsula, Alaska, USA. After treatment with rotenone in 2009 to exterminate an invasive fish species, 3047 anadromous Threespine Stickleback from a spawning run in Rabbit Slough (62.6893 N, 149.427E) in the Matanuska-Susitna Borough, Alaska, were released into the lake in the summer of 2011 (Fig. 1A). Tests with trout late in the summer of 2009 indicated that fish did not survive the rotenone treatment (Supplementary Note 1). The lake is isolated from other bodies of water, and sticklebacks (and other fish) were not detected by exhaustive sampling after the rotenone poisoning. Stickleback samples have been collected annually from this lake since the year after its founding in 2011 (i.e., from 2012) for phenotypic and genetic analysis[12,21]. In this study, we focused on samples collected in 2013, 2014, 2015, 2017, and 2020, and we referred to these samples as SC2013, SC2014, SC2015, SC2017, and SC2020, respectively. These samples correspond to 2, 3, 4, 6, and 9 years after the founding of the new Scout Lake population. In addition, we included two samples from the anadromous source population, Rabbit

Slough, collected in 2009 and 2019. These two timepoints from the Rabbit Slough represent the genomic diversity in the ancestral population over the same period as examined in Scout Lake. We denote these two Rabbit Slough samples as RS2009 and RS2019. Despite being collected ten years apart, the two Rabbit Slough samples showed highly correlated allele frequencies (Figs. S1 and S2) and highly consistent results when used as baselines for inferring processes in the ancestral oceanic population.

The adults stocked in Scout Lake in 2011 produced offspring after release but did not survive the winter (Fig. 1B and Fig. 3 from Kurz et al.[22]). The F1 progeny likely did not successfully reproduce until 2013. However, small sexually mature males and females were captured in 2012 (Fig. 1B–D; Kurtz et al.[22]). This finding indicates that the F1 survived in the lake for two years, which is not surprising given that F1 offspring of anadromous Threespine Stickleback typically spend part of the earliest and reproductive segments of their life cycle in freshwater[23,24] and thus are physiologically capable of tolerating freshwater. These phenotypic results show that SC2014 represents one-year-old F2 offspring of the F1 generation (Fig. 1E), as any stickleback born in the sampling year would have been too small to be captured with the sampling methods employed (i.e. minnow traps). Thus, the first generation of offspring of the parents that were born and spent their entire life in the lake were adults captured in 2014.

### Jackpot carriers increased in frequency in Scout Lake

To infer whether haplotypes with numerous freshwater-adaptive alleles were assembled after introduction to the lake or were already present in rare large haploblocks in a few of the introduced anadromous founders, we used a TN5 transposase-based approach to sequence 432 genomes at coverages ranging from 0.43X to 1.84X (low-coverage). There were 336 low-coverage genomes from individuals collected from two ($n = 96$), three ($n = 48$), four ($n = 96$) and six ($n = 96$) years after the Scout Lake population was founded, and 96 genomes from RS2019. We sequenced additional 20 genomes from SC2020 at a mean of 25.7X coverage (high-coverage) (Methods, Supplementary Data 1), and also included 20 previously published high-coverage genomes from RS2009[12].

We defined multi-SNP haplotypes for each of 344 loci with freshwater-adaptive alleles that we previously identified to experience rapid and significant frequency increases in three lake stickleback populations from Cook Inlet, Alaska[12,25]. These populations, which included the Scout Lake population, were founded by anadromous ancestors. The 344 loci range from a few base pairs to kilobases, with a median size of 27.3 kb. Each multi-SNP haplotype consists of the most significant freshwater-adaptive SNP within the locus and selected neighboring SNPs whose freshwater allele frequencies are highly correlated ($r > =0.99$) across our time-series data from the three Threespine Stickleback populations[12]. We excluded all loci with less than three SNPs, leaving 300 haplotypes that contain three to 3658 SNPs.

We then developed an approximate genotype likelihood approach suitable to call the diploid state of freshwater-adaptive loci as homozygous oceanic, heterozygous, or homozygous freshwater from low-coverage genomic data (Methods). We validated this method by applying it to experimental crosses between parents with known oceanic and freshwater ecotypes derived from a lake (60.914517°, −149.101545°) at mile 87 of Seward Highway a few kilometers east of Girdwood, Alaska (Methods). Genotypic states at loci inferred using our likelihood method were consistent with expectations from our crosses (Fig. S3). Parents were scored as homozygous for the alleles associated with their ecotype at most loci. More importantly, the offspring of parents with alternative homozygous states for a locus were heterozygous at the called locus (Methods, Fig. S3). We then applied this approach to the whole genomes from Rabbit Slough (2009 and 2019) and Scout (2013–2020). Loci with missing genotype calls in any

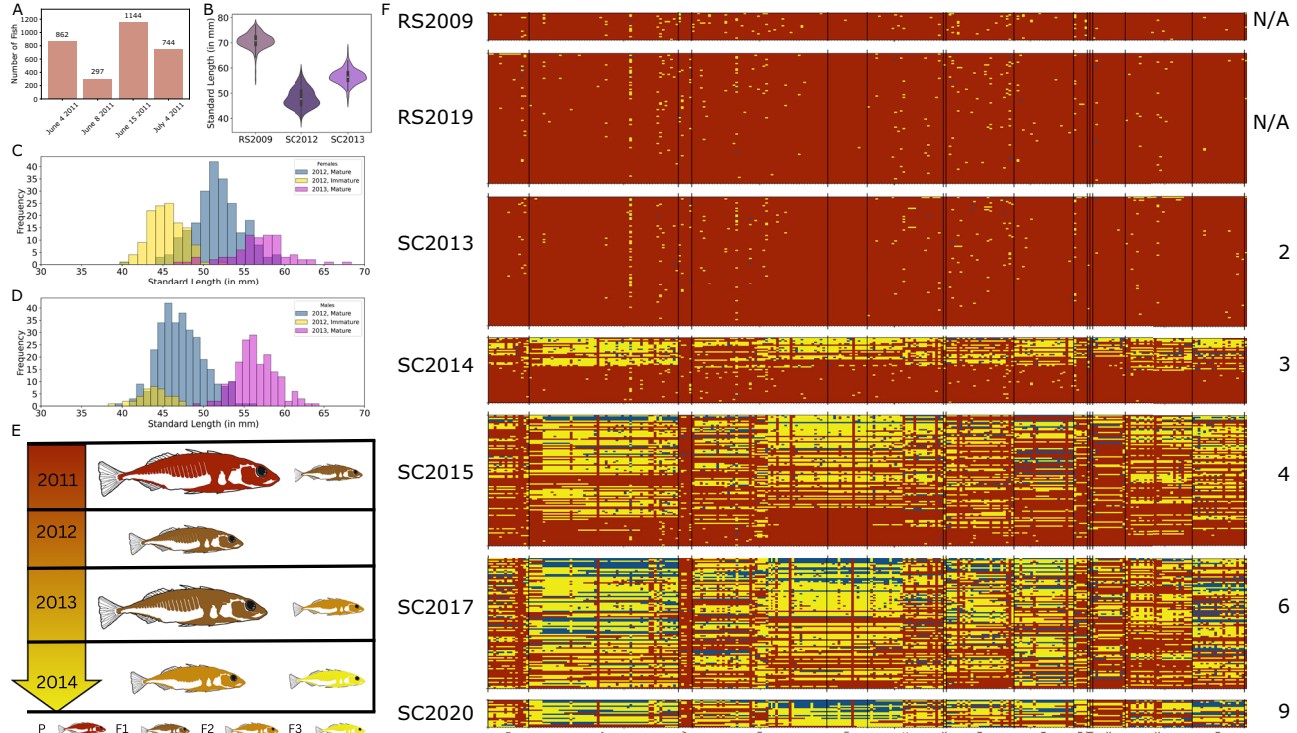

**Fig. 1 | Genotypes of freshwater adaptive loci. A** Number of anadromous Rabbit Slough used to found Scout Lake in 2011. **B** Standard length (SL) of Rabbit Slough 2009, SC2012 and SC2013 samples. The SL is a measure of the size of the fish. Sample sizes, *n*: RS2009 = 214, SC2012 = 709, SC2013 = 254. **C** Sexual maturity of females in SC2012 and SC2013 samples. **D** Same as (**C**) but for males. **E** Inferred generations based on (**B–D**) for the samples collected in the first three years after founding of Scout Lake. **F** Genotypes of individuals used in this study at freshwater adaptive loci. Each column is a multi-SNP locus, and each row is an individual genome. Each multi-SNP haplotype consists of the most significant freshwater-adaptive SNP within each of 344 loci identified in Roberts Kingman et al. 2021[12] and selected neighboring SNPs whose freshwater allele frequencies are highly correlated (*r* > =0.99) across time-series data from three Threespine Stickleback populations[12]. N/A years since founding indicates specimens collected in Rabbit Slough, representative of the anadromous founders of Scout Lake stickleback population; 2,3,4,6 and 9 years since founding are samples in Scout Lake collected in 2013, 2014, 2015, 2017, and 2020, respectively. There are 96 specimens in RS2019, SC2013, SC2015 and SC2017 samples; 47 and 20 in SC2014 and SC2020 samples respectively. Each locus has 3 to 3658 SNPs. For (**B** and **C**), the bars have three colors: yellow, light blue, and purple. The other colors result from overlap of these colors. Fish images in panel E have been modified from Roberts Kingman, Garrett A., et al., Science Advances, DOI: 10.1126/sciadv.abg5285 (2021), AAAS, with permission. © The Authors, exclusive licensee to the American Association for the Advancement of Science.

individual were excluded, resulting in 280 out of the original 344 loci where all individuals in the study had their genotypes called.

Freshwater-adaptive loci were almost always homozygous for oceanic alleles in the samples from the ancestral population and SC2013 (2 years since founding) (Fig. 1F). Individuals in these samples were either completely homozygous for oceanic alleles (0 out of 20 RS2009; 12 and 15 out of 96 specimens for RS2019 and SC2013, respectively) or possessed freshwater-adaptive alleles at only a few loci (average number of freshwater-adaptive alleles in RS2019 = 2.1 [range: 0–9], SC2013 = 2.25 [range: 0–30], Fig. 1F). The mean proportion of freshwater adaptive alleles in the 20 RS2009 individuals were slightly higher than in either the RS2019 and SC2013 samples (RS2009 = 0.0122 vs RS2019 = 0.00569: $t = 5.7224$, $p = 0.000008$, RS2009 vs SC2013 = 0.005952: $t = 4.9428$, $p = 0.000020$). The mean proportion of freshwater-adaptive alleles did not differ significantly between the RS2019 and SC2013 samples (RS2019 = 0.00569 vs SC2013 = 0.005952: $t = -0.3475$, $p = 0.728697$; Figs. 1F, 2A) or RS2009 and SC2013 samples (RS2009 = 0.000875, SC2013 = 0.005952: $t = -1.7967$, $p = 0.0732$, Fig. 2A).

We estimated the size of a contiguous block of freshwater-adaptive alleles by the genome coordinates that defined SNPs. For a series of adjacent loci with freshwater-adaptive alleles, we estimated the size of the block by calculating the difference between the first SNP of the first locus with a freshwater-adaptive allele and the last SNP of that series. For the results presented below, we did not impose a maximum threshold distance between adjacent loci when defining a haploblock. However, we tested the impact of imposing a maximum threshold, and this criteria did not change the overall interpretation of the haploblock distributions ("Methods", Figs. S4–S5). Of the 81 individuals in SC2013 with freshwater-adaptive alleles, only seven had two or more loci forming a contiguous block with freshwater-adaptive alleles, with blocks of alleles spanning between 0.17 Mb to 3.86 Mb [0.41 cM to 4.57 cM] (Fig. 3A, Supplementary Data 2).

In stark contrast to the SC2013 sample, in the third year after founding (SC2014, F2 generation, Fig. 1E), nearly half of the individuals in our sample possessed large numbers of freshwater-adaptive alleles (Figs. 1F, 2B, A). The mean proportion of freshwater-adaptive alleles in SC2014 was significantly different from that of SC2013 (mean proportion of freshwater alleles in SC2014 = 0.1222 vs SC2013 = 0.0048; $t = 6.3156$, $p = 8.923e-08$; Figs. 3B, 2A). The distribution of the number of freshwater-adaptive alleles per individual in the SC2014 sample was bimodal (Fig. 2A), which suggested that two distinct groups were in the SC2014 sample (Fig. S6). One group of individuals appears to possess a few number of freshwater-adaptive alleles similar to the ancestral population and the SC2013 samples, while the other had large numbers of freshwater-adaptive alleles (Fig. 2A).

Therefore, to aid our understanding of genomic change during adaptation to freshwater, we categorized individuals as jackpot carriers and non-jackpot individuals using Bayesian Gaussian Mixture Modeling (BGMM)(Methods, Fig. 2B). The BGMM categorized

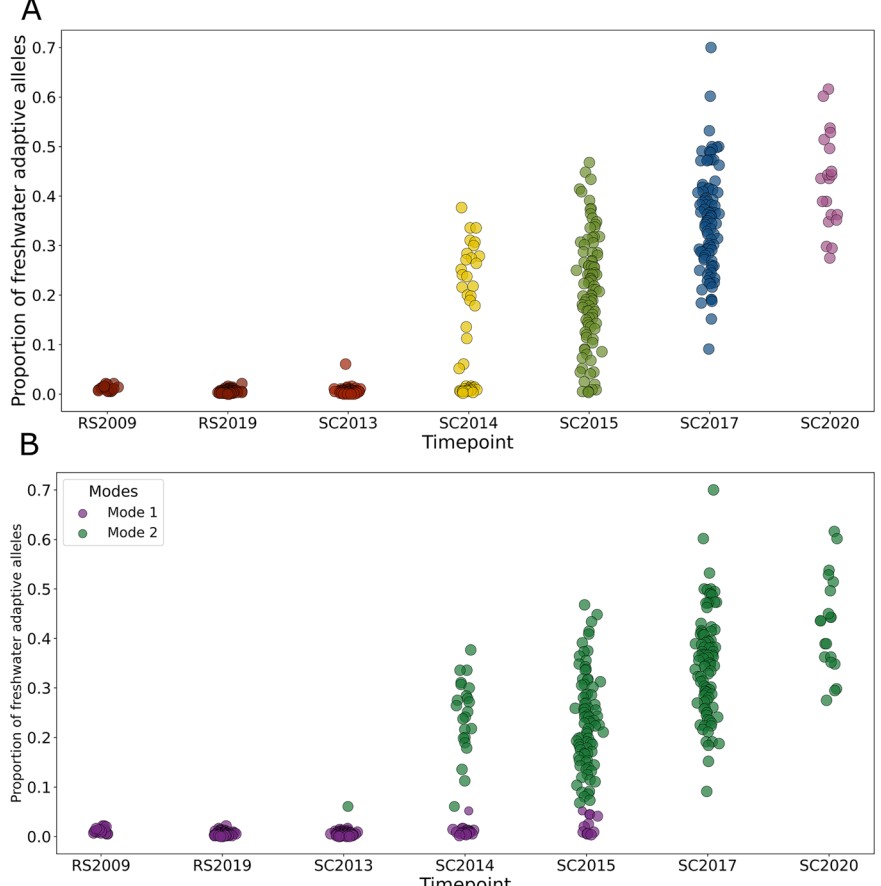

**Fig. 2 | Classification of individuals as jackpot carriers. A** Quantification of the freshwater content for each sampled timepoint. The proportion of freshwater adaptive alleles for each time point was estimated by counting the number of freshwater alleles at called loci (out of 280 adaptive loci without missing data). **B** Categorization of individuals based on the content of freshwater adaptive alleles at adaptive loci. Individuals were categorized as non-jackpot individuals (mode 1) if they had less than 5% of freshwater adaptive alleles and jackpot carriers otherwise, using Bayesian Gaussian Mixture Modelling (BGMM).

individuals with 5% freshwater adaptive alleles as jackpot carriers with a posterior probability of 0.95 and those with 8% freshwater adaptive alleles were classified as jackpot carriers with 100% certainty (Fig. S7). Based on the BGMM classification, there were no jackpot carriers in RS2009 and RS2019 samples and one jackpot carrier in SC2013 sample (i.e., 1%). There were 23 out of 48 in SC2014 (i.e., 48%), 84 out of 96 in SC2015 (i.e., 88%), and all in SC2017 and SC2020 (i.e., 100%, Supplementary Data 1). We observed that 27 out of the 47 individuals in SC2014 had contiguous blocks with freshwater-adaptive alleles. These contiguous blocks were mostly heterozygous (Fig. S8) and spanned 0.07 Mb [0.044 cM] to 22.33 Mb [70.48 cM], with a mean of 2.40 Mb [2.9 cM] (Figs. 3A and 3B, Supplementary Data 2).

Threespine Stickleback in the Cook Inlet lakes become reproductively mature at around one to two years old[26], and results from Kurz et al.[22] indicated that by 2014, the captured individuals were F2 (Fig. 1E). Therefore, the observed jackpot carriers are likely not products of re-assembly via recombination from anadromous founders which possessed only a few freshwater-adaptive alleles. Jackpot carriers must have been present in the founding population, but at such low frequencies that they were undetectable in our limited sampling from the ancestral population (RS2009 and RS2019) or in the SC2013 sample. Based on previous estimates of the frequency of individuals that can be classified as jackpot carriers in ancestral marine environments of 0.1%[12,17], the probability of including at least one jackpot carrier in a sample of 96 individuals is only 9% (assuming binomial sampling probabilities). The rapid increase in the frequency of jackpot carriers from 1% in the F1 generation (i.e., SC2013), which

was born and spent its entire life in the lake, to almost 48% in the F2 generation (i.e., SC2014) was likely a result of their markedly greater fitness in the freshwater environment compared to individuals with few freshwater-adaptive alleles.

Low survival and/or low reproductive success of the vast majority of non-jackpot carriers during years two and three after founding resulted in a population bottleneck that was manifested by a markedly reduced catch per unit effort (CPUE), a relative measure of abundance (Fig. 4A). In the SC2013 sample, the decline in CPUE was consistent with a decrease in genome-wide genetic diversity compared to the ancestral population samples (Watterson's $\Theta$ = 0.005328 for RS2009 and $\Theta$ = 0.00535 for RS2019 vs 0.0048 for SC2013, Fig. 4B). CPUE declined further in SC2014, and a correspondingly more significant reduction in genetic diversity of almost 50% compared to the ancestral population ($\Theta$ = 0.00359 for SC2014, Fig. 4B). This population decline in Scout Lake caused Tajima's D to become increasingly positive in the SC2013 ($D$ = 0.263) and the SC2014 ($D$ = 1.618) sample despite the ancestral population having a negative value ($D$ = −0.722 for RS2019) that likely reflects a population expansion in the ancestral environment. The genome-wide site frequency spectrum (SFS), estimated using ANGSD[27], indicates progressive, genome-wide loss of singletons, with a particularly drastic distortion of the SFS in SC2014 for rare alleles, and is consistent with the dramatic population decline within the Scout Lake population (Fig. 4C). The SFS between the high-coverage RS2009 and SC2020 (Fig. S9) samples also showed loss of singletons in the Scout Lake persisting through several generations after founding.

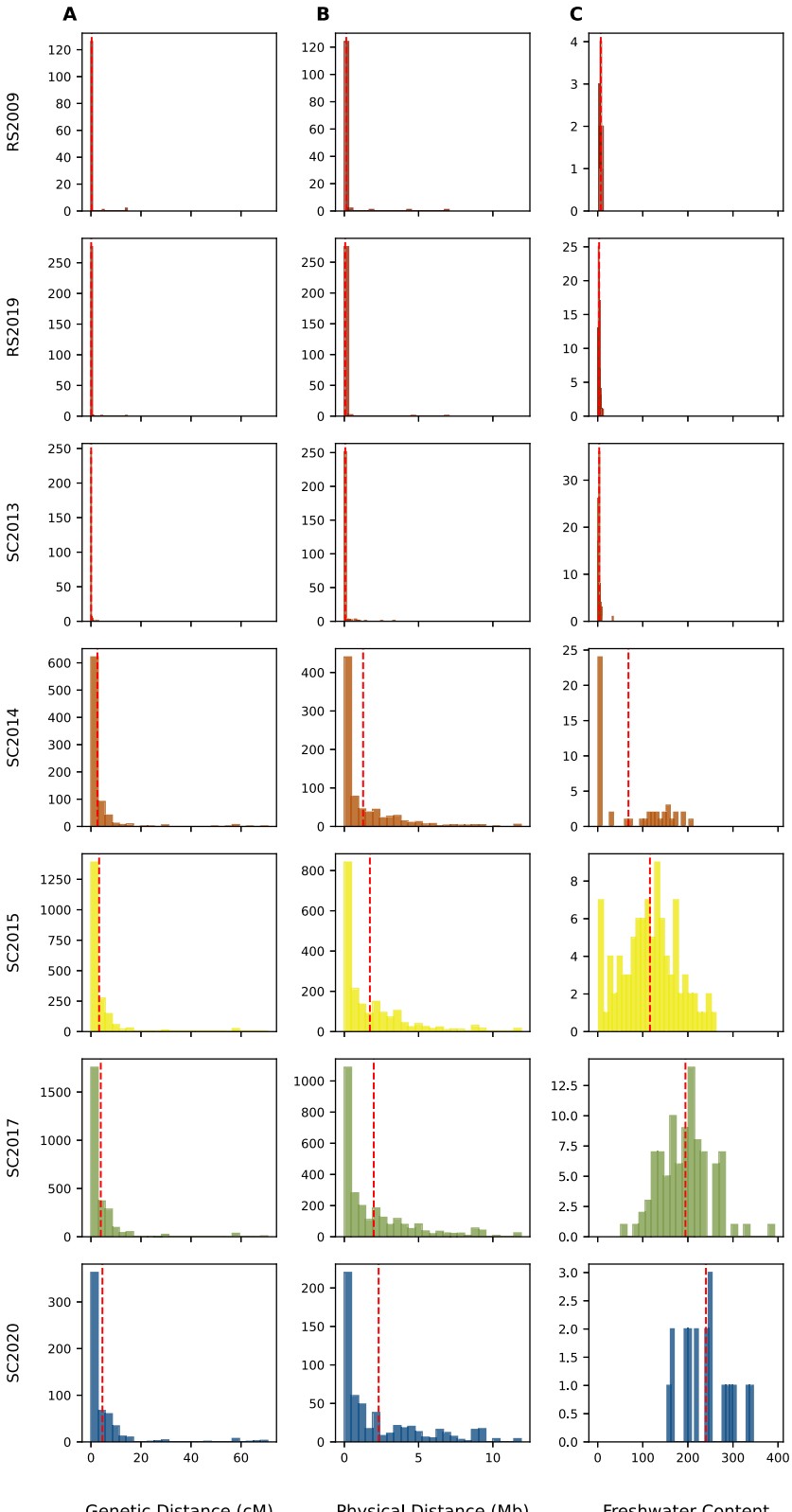

**Fig. 3 | Distance between freshwater adaptive loci and freshwater content at each locus during early stages of freshwater adaptation. A** Distribution of measured genetic distance across the timepoints. **B** Distribution of measured physical distance across the timepoints. **C** Freshwater content is estimated as the proportion of freshwater dosage of all loci in a time point. In (**A** and **B**), a contiguous block is defined as two or more consecutive loci with freshwater adaptive alleles. The genetic and physical distances are estimated from the first SNP of the first locus and last SNP of the last locus of a contiguous block. In (**C**), a locus with homozygous freshwater has a dosage of 2, heterozygote has 1, and marine homozygote has 0. The red dashed line is the mean for each plot.

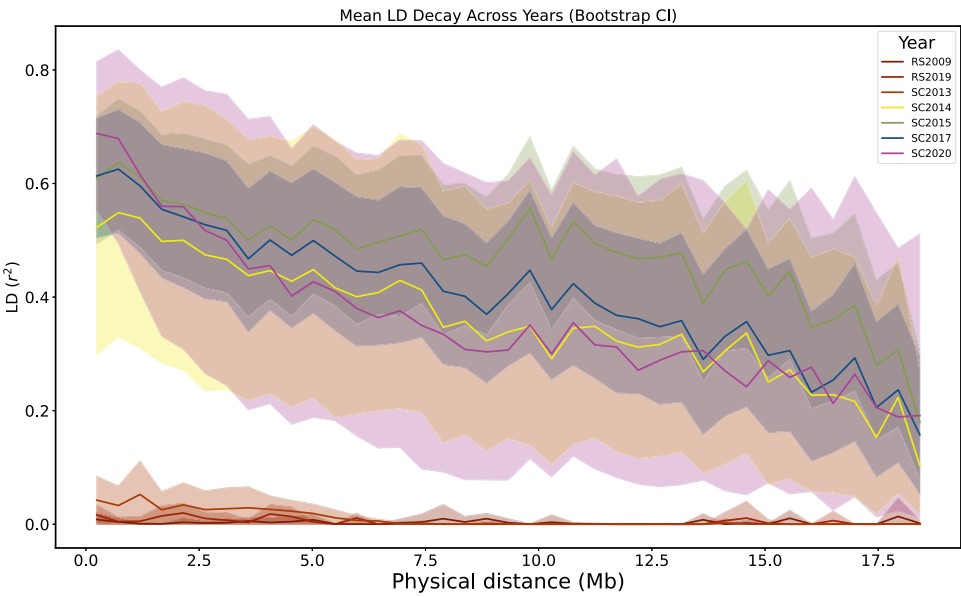

**Fig. 4 | Linkage disequilibrium (LD) decay at freshwater adaptive loci.** The LD as measured by the $r^2$ was estimated using hard-called locus-based genotypes, coded as 0 for homozygous freshwater, 1 for heterozygous, and 2 for homozygous marine. Bootstrapping was used to quantify the uncertainty around the $r^2$ estimates.

After this bottleneck, the CPUE increased, and there was a relative increase in the frequency of rarer variants in the SFS for the SC2015 and SC2017 samples compared to SC2014. The number of freshwater-adaptive alleles increased consistently over time in samples collected after the bottleneck; the mean proportion of freshwater-adaptive alleles in SC2015, SC2017 and SC2020, were 0.2078, 0.3504, 0.4292, respectively (Figs. 1F, 2B, A). However, the size of freshwater haplo-blocks, as measured by the haploblock length in physical base pairs or centiMorgans (Fig. 3), did not change at the same pace, as would have been expected if recombination had been bringing freshwater-adaptive alleles together from multiple parents each generation to form increasingly larger haploblocks.

There was no significant difference in the mean haploblock lengths between the two Rabbit Slough samples (RS2009 vs RS2019; $t(158.42) = 1.2021$, $p = 2.311e\text{-}01$), as well as between the RS2019 and SC2013 ($t(334.40) = 0.4335$, $p = 6.645e\text{-}01$; Figs. 3, S4, S5). There was a statistically significant difference between the mean haploblock lengths of SC2013 and SC2014 ($t(802.17) = -9.7188$, $p = 3.4958e\text{-}21$), SC2014 and SC2015 samples ($t(1515.87) = -2.2363$, $p = 2.5478e\text{-}02$) and SC2015 and SC2017($t(4475.41) = -2.7869$, $p = 5.3432e\text{-}03$). There was no statistically significant difference between the mean haplo-block lengths of SC2017 and SC2020 ($t(748.07) = -1.4509$, $p = 1.4724e\text{-}01$, Fig. 3). We further explored the differences in haplo-block lengths using Cohen's D, which expresses the mean difference relative to the pooled standard deviation. We estimated Cohen's D and bootstrapped confidence interval for each pair of consecutive timepoints. There was a small difference between RS2009 and RS2019 ($D = -0.1585[-0.352$ to $0.087]$) and RS2019 and SC2013 ($D = 0.0044[-0.1341$ to $0.26]$) compared to SC2013 and SC2014 ($D = 0.4025[0.367 - 0.4510]$). Between SC2014 and SC2015, there was a small difference in the mean haploblock lengths ($D = 0.093[0.012$ to $0.172]$), suggesting that despite the statistically significant t-test, the difference was not substantial at this time period. We observed similar Cohen's D between mean haploblock means between SC2015 and SC2017 ($D = 0.083[0.024$ to $0.138]$) as well as SC2017 and SC2020 ($D = 0.074[-0.025$ to $0.174]$). These results suggest that there was limited increase in the haploblock lengths across the time periods, especially after the apparent demographic bottleneck in 2014, which would be expected if the haploblocks were being reassembled by recombination.

Venu et al.[19] indicated that as many as half of all chromosomes in males, as well as a third of chromosomes in females, could be inherited without any recombination event per generation. In addition, recombination is suppressed in individuals that are heterozygous for marine and freshwater haplotypes at adaptive loci. Our sampling period covered nine years, spanning 5-6 generations, and the majority of the individuals sampled after the bottleneck were predominantly hetero-zygous at freshwater adaptive loci. We estimated linkage dis-equilibrium (LD) across freshwater adaptive loci (Methods) to assess whether recombination had any substantial effect on the integrity of haploblocks. We observed an elevated LD from SC2014 decay compared to SC2013 and samples from the ancestral population, con-sistent with a strong directional positive selection increasing the frequency of adaptive haploblocks or potentially the effect of demo-graphic bottleneck that occurred in 2014 (Fig. 5). In subsequent years, we found no observable patterns of LD decay, suggesting that there was not enough time for recombination to have substantial effect on the integrity of haploblocks.

The homozygosity of freshwater-adaptive loci also increased over time from SC2014 (Fig. S8), which could reflect the mating between heterozygous jackpot carriers. These results demonstrate that haplo-blocks of freshwater-adaptive alleles were not re-assembled through progressive recombination over many generations, but instead the frequencies of freshwater-adaptive alleles increased after the F1 gen-eration by matings predominantly between the descendants of jackpot carriers present in the founders. These jackpot carriers likely had an increased fitness in the freshwater environment.

**Biological kinship and inbreeding during freshwater adaptation**
Our sequence data from multiple whole genomes during the earlier years after the Scout Lake population was founded provided a unique opportunity to explore the genealogical relationships among indivi-duals as the population adapted to the new freshwater environment and to infer the dynamics of haploblock inheritance. We estimated biological relatedness among our low-coverage genomes using READv2[28,29] and ngsRelate[30]. The estimates of relatedness coefficients from the two methods were highly correlated ($r > 0.92$ for all time points; Fig. S10). However, in situations in which there is increased allelic sharing compared to expectations from panmixia due to inbreeding, we might expect READv2 relatedness coefficients to be

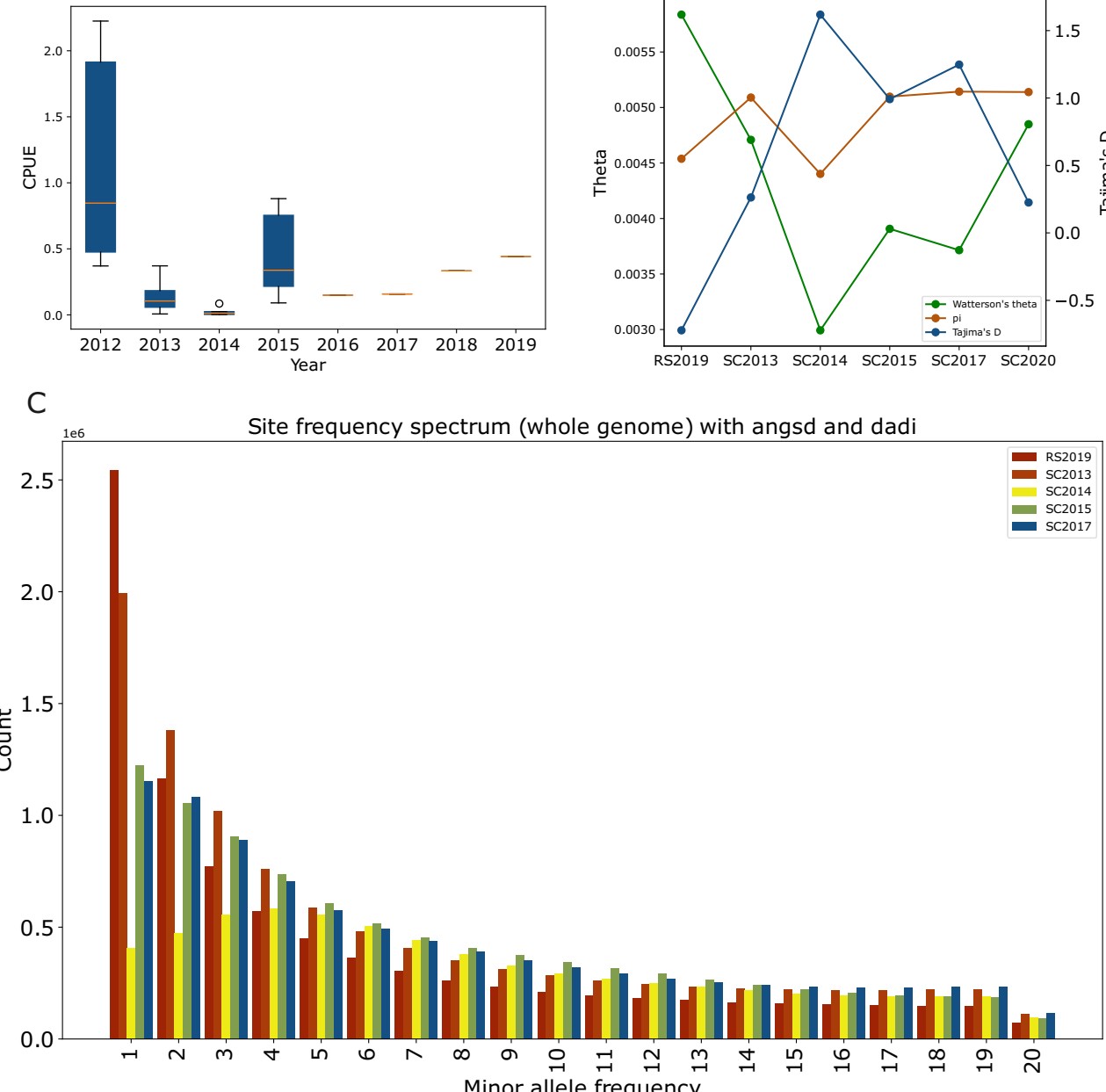

**Fig. 5 | Measures of demographic changes during rapid adaptation in Scout Lake. A** Catch per unit effort (CPUE). CPUE was measured as the number of sticklebacks caught per trap hour. For the initial study period (2012–2015), mean CPUE was calculated from multiple replicates: $n$ for 2012 = 6, 2013 = 13, 2014 = 6, 2015 = 7. While data for the 2016–2019 period represent single observations. **B** Summary statistics (Tajima's D, theta and nucleotide diversity) calculated from the whole genome for Rabbit Slough, SC2013, SC2014, SC2015, SC2017 and SC2020. **C** Site Frequency spectrum (SFS) estimated from the whole genome. The SFS was folded in dadi and projected down to 40. For clarity, we only show the first 20 polymorphic sites in the SFS. We excluded SC2020 from the SFS in (**C**) because of the differences in depth of coverage with the other samples. The SFS with the SC2020 sample can be found in Fig. S9.

more robust as the approach relies on observed rates of allele mismatching, while ngsRelate utilizes allele frequencies under the assumption of Hardy-Weinberg equilibrium, leading to potentially inflated values of relatedness (Methods). Therefore, we report the results based on READv2, which identifies up to third-degree relatives. All pairwise relationships detected using READv2 and NGSrelate are provided in Supplementary Data 4.

We also estimated individual inbreeding coefficients, $F$, with ngsRelate, which can estimate inbreeding from low-coverage genomes (up to approximately 4×[30]). A down-sampling experiment of the high-coverage genomes from the SC2020 sample demonstrated that

ngsRelate underestimates inbreeding coefficients from low-coverage of 1X by a factor of ~50% compared to high-coverage (~30X) whole genome data ("Methods", Fig. S11); thus, the reported inbreeding coefficients from our low-coverage genomes from the SC2013 to SC2017 samples are likely underestimated.

There were no close biological kinships (up to third-degree relatedness) in samples taken from the ancestral population, suggesting that the founders of the Scout Lake population were drawn from an outbred anadromous population with a large effective size ($N_e$). Despite possessing identical proportions of freshwater-adaptive alleles as the ancestral population (Fig. 1F), the SC2013 sample had two

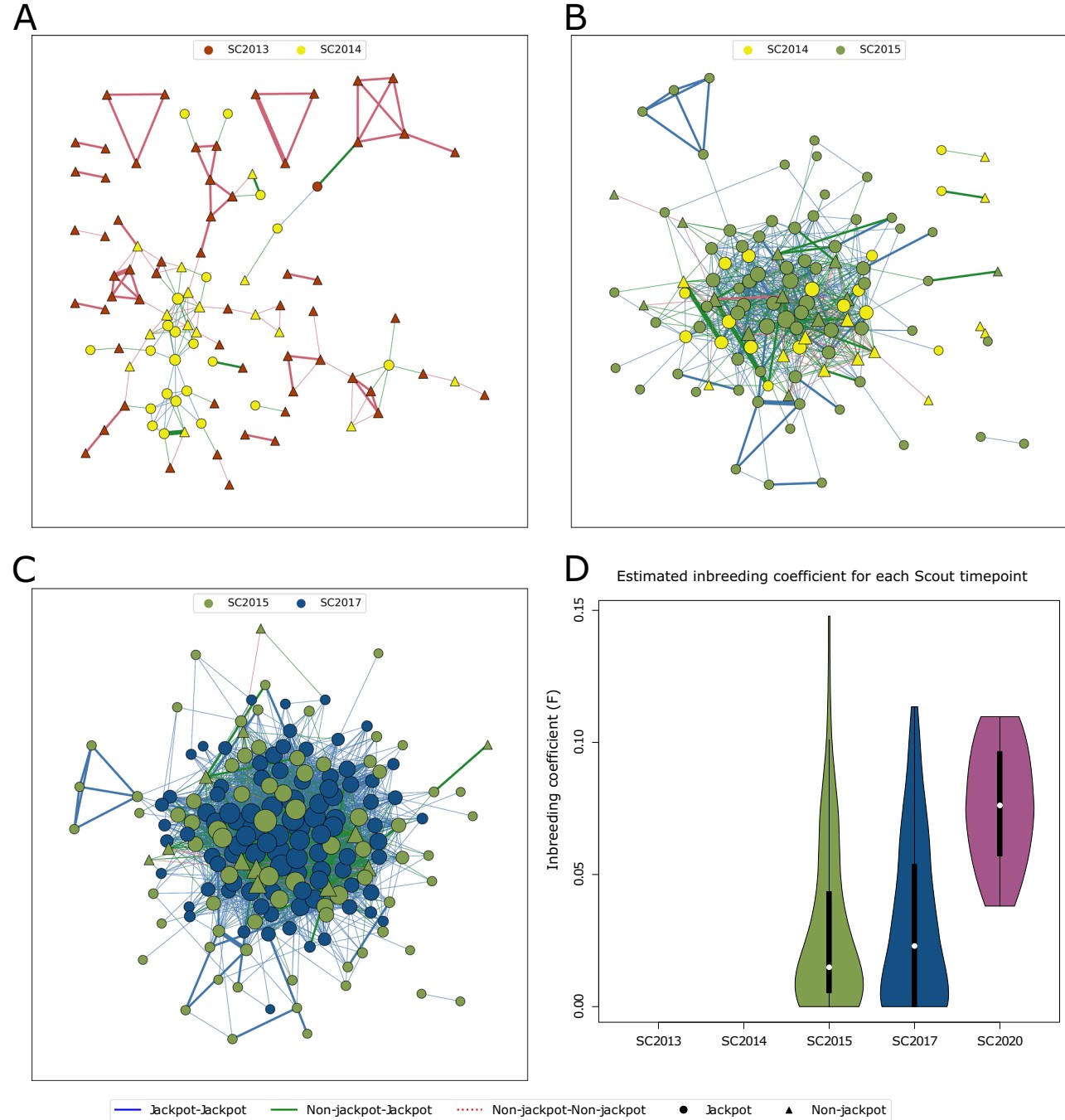

**Fig. 6 | Inter-year biological relationships and inbreeding. A** Relatedness between SC2013 (2 years since founding) and SC2014 (3 years since founding). **B** Relatedness between SC2014 (3 years since founding) and SC2015 (4 years since founding). **C** Relatedness between samples SC2015 (4 years since founding) and SC2017 (6 years since founding). **D** Inbreeding coefficients estimated from ngsRelate at the various time points. No inbred individual was observed in SC2013 and SC2014, 2 and 3 years after founding. In (**A–C**), the thickness of edge reflects the degree of relatedness between the nodes: the thickest edge first-degree, intermediate second-degree and the thinnest edge third-degree relatives. The size of the node increases with increasing number of relatives. In (**D**), the sample sizes for SC2013 = 96, SC2014 = 48, SC2015 = 96, SC2017 = 96, SC2020 = 20.

first-, 38 second- and five third-degree relatives (Figs. 6A, S12A), indicating that this population is descended from a small number of fish with a much more limited choice of mates than the ancestral anadromous population.

Some related pairs involved one individual from the SC2013 sample and the other from the SC2014 sample. There were 36 third-degree and two second-degree related pairs between these two time points (Fig. 6A). Some related pairs had one of the pair being a jackpot carrier in SC2014 and the other being a non-jackpot individual in

SC2013, suggesting that jackpot carriers mated with non-jackpot individuals early on in the colonization process.

While the majority of relatedness estimates within the SC2013 sample (Fig. S12A) formed small networks, each containing only a few individuals, out of a total of 60 related pairs within 2014, 57 were connected within a single large network (Fig. S12B). The one first-degree relationship was a sibling pair. Apart from four relationships between jackpot and non-jackpot individuals, this network's remaining pairs were all jackpot carriers (Figs. 6A, S12B). Most jackpot carriers (20

out of 23 individuals) identified in the SC2014 sample were connected within this single network. The third-degree relationships of jackpot carriers sampled in 2014 are likely to be cousins or half-avuncular assuming the typical lifespan of freshwater Alaskan Threespine Stickleback is one to two years[31].

We observed one first-degree (sibling pair), nine second-degree, and 315 third-degree pairs of relatives between specimens from the SC2014 and SC2015 samples. Within the SC2015 sample (Fig. S12C), there was one first-degree (sibling pair), 36 second- and 298 third-degree pairs. These 335 connections were between 77 out of 96 sampled individuals in SC2015, 55 of which were jackpot carriers (there were 78 jackpot carriers in SC2015). Most individuals sampled in SC2014 and SC2015 were members of the large network of relatedness which contained most of the sampled jackpot carriers (20 out of 23 in SC2014 and 67 out of 78 jackpot carriers in SC2015; Fig. 6B). Unlike in the SC2013 and SC2014 samples, some individuals in SC2015 showed evidence of inbreeding (Fig. 6D), with $F$ values ranging from 0.0 to 0.149 (mean=0.0289). Thus, the jackpot carriers observed in SC2015 were mostly related to the SC2014 jackpot carriers, suggesting that the SC2015 jackpot carriers descended from the network of jackpot carriers in SC2014.

The network of relationships between the SC2015 and SC2017 sample provided further evidence that population growth was driven by jackpot carriers, with 34 second- and 1201 third-degree pairs in 167 out of 192 (i.e. 87%) individuals from SC2015 and SC2017. All relationships, except seven third-degree and one second-degree relative, were between jackpot carriers. By 2017, all sampled individuals were jackpot carriers, and 93 out of them were in a single network (Fig. S12D). The individuals sampled in SC2017 had an average inbreeding coefficient of 0.0316 (range [0.0, 0.1121]). None of the 20 individuals in the SC2020 sample were first-, second- or third-degree relatives, though we identified three third-degree relatives between SC2017 and SC2020 individuals (Supplementary Data 4). To test if this lack of relatedness within SC2020 was significantly lower than in SC2017 or simply a result of the smaller sample size (SC2017 $n = 96$, vs. SC2020 $n = 20$), we performed a permutation test by randomly sampling 20 individuals from the SC2017 sampled 1000 times and observing the number of pairwise relatives of at least third-degree. In no permutation did we observe 20 random individuals from SC2017 with no relatedness (mean = 57.2, min = 16, max=112, Fig. S13). This result suggests that the lack of relatedness in the SC2020 sample was likely to be a result of population growth that reduced the probability of sampling relatives and not a result of the small sample size. Despite the deficiency of related pairs, individuals in SC2020 were, on average, inbred with an $F$ of 0.078 (range [0.038, 0.110]). The dearth of relatives within SC2020 suggests that the population had become large enough to allow more mate choice by this time, with the increased $F$ reflecting the legacy of inbreeding from the preceding generations.

## Patterns of polymorphism, selection and genetic load

When a population experiences a decline in size, the increased probability of inbreeding can increase genetic load[32–34] and reduce mean population fitness[34,35]. Rare, deleterious, recessive alleles previously masked in heterozygotes become exposed as homozygotes and reduce individual fitness[32,33,36]. To determine how the dynamics of the observed bottleneck affected the genetic load of the Scout Lake population, we estimated the site frequency spectrum (SFS) genome-wide as well as at 4-fold (silent) and 0-fold (amino acid replacement) degenerate sites for samples from the time series (Methods).

The proportion of rare alleles in an SFS provides insight into the demographic history of a population, with excess rare alleles suggesting population growth, while a deficit may suggest population decline. This pattern results from each individual introducing unique mutations to the population as the population grows. Demographic processes affect both 0- and 4-fold sites equally, whereas natural selection is more likely to impact 0-fold sites. As expected, 0-fold sites generally demonstrate approximately a third of the diversity of 4-fold and genome-wide sites for all samples, presumably due to the effects of purifying selection (Fig. 7H). The ancestral population (RS2019) showed a slight excess of singletons at 0-fold sites compared to 4-fold and genome-wide sites, possibly due to the segregation of slightly deleterious recessive alleles in the ancestral anadromous population at low frequencies (Figs. 7A, S14A). During the bottleneck from 2013 to 2014 (SC2013 and SC2014 samples), there was a similar reduction in Watterson's $\Theta$ at 0-fold and 4-fold sites and genome-wide (30%, 34% and 25%, respectively; Fig. 7D, E), indicating that a demographic factor, the population bottleneck, shaped most of the patterns of diversity during this time (Fig. 7H).

However, within the SC2014 sample a notable increase in singletons at 0-fold sites relative to that expected at genome-wide sites was detected, with 4-fold sites being intermediate, presumably reflecting linkage to 0-fold sites (Fig. 7D). Compared to non-jackpot individuals, the SFS of jackpot carriers was skewed towards more common variants with no significant increase in 0-fold singletons, likely reflecting the close genealogical relationships between these individuals (Fig. S15). The singletons in non-jackpot carriers reflected the excess of singletons at 0-fold sites in the adult F2 generation (SC2014). We could, therefore, attribute the observed singletons at 0-fold sites almost entirely to non-jackpot carriers. However, this excess of singletons at 0-fold sites was not observed in the samples collected from 2015, 2017, and 2020 (SC2015, SC2017, and SC2020). As described above, we also observed high levels of relatedness and inbreeding in SC2015 and SC2017 as shown by increased homozygosity (Fig. S8). Thus, slightly deleterious 0-fold sites were likely being purged from the population as a result of a) non-jackpot individuals failing to leave descendants except when they mated with jackpot carriers and b) an increased chance of recessive deleterious alleles being homozygous and thus being selected against.

## Testing jackpot-mediated rapid adaptation with forward-in-time simulations

To further explore whether our observed data at Scout Lake could be better explained by the inheritance of large haploblocks of freshwater adaptive alleles, rather than their gradual accumulation through recombination, we employed forward-in-time Wright-Fisher simulations in SLiM[37] to model the process of freshwater adaptation. Our simulations closely follow the scenarios in Galloway et al. 2020[18] and Roberts Kingman et al. 2021[12] (Supplementary Note 4, Fig. S16), with the latter using a machine learning inference framework to identify a selective regime that accurately fits observed allele frequency trajectories at adaptive loci across multiple rapidly adapting freshwater populations, including Scout. Based on the demographic and selective parameters inferred in Roberts Kingman et al. 2021, we let the simulation run for 1000 generations, which was enough time for the system to reach equilibrium, and then founded a new freshwater population under two scenarios: A) a scenario where both jackpot individuals (maintaining the definition of 5% freshwater alleles) and non-jackpot carriers are present in the founding population of Scout Lake and B) a scenario where only non-jackpot carriers are present in the founding population of Scout Lake.

We estimated the total freshwater adaptive alleles in the founding population under both scenarios (Fig. 8A) and found that the freshwater content was not significantly different (KS test: 0.08, $p$-value:0.908). Thus, the presence of a small number of jackpot individuals in the founding population (Fig. S17) did not skew the overall content of freshwater alleles available for adaptation for scenario A compared to B. We then calculated the proportion of freshwater adaptive alleles in the generation corresponding to the observed sampled timepoints from Scout Lake. We found that we

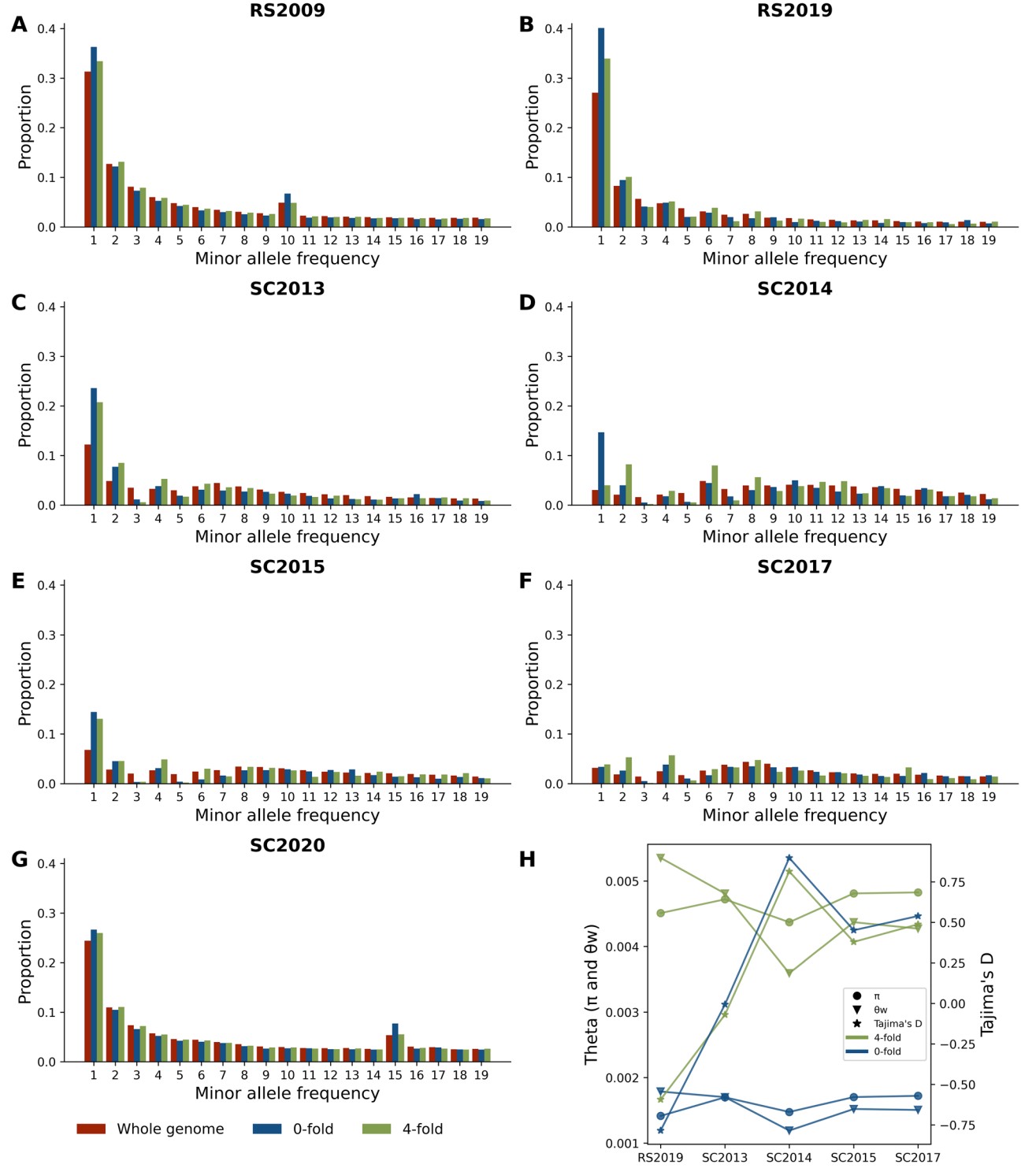

**Fig. 7 | Estimate of selection and demography. A–G** Site Frequency Spectrum (SFS) at the whole genome, 4-fold and 0-fold degenerate sites at various time points. **H** Estimates of theta and Tajima's D for the various time points at 4-fold and 0-fold sites.

could only replicate the observed mean proportions of freshwater adaptive alleles in simulations under scenario A (Fig. 8C), demonstrating that jackpot individuals are needed in the founding population for the observed rapid adaptation in Scout Lake. When jackpot carriers were removed from the founding population, rapid adaptation was not possible in any simulation (Fig. 8B). Thus, our simulations indicate that it is crucial to have individuals with large blocks of freshwater adaptive alleles for rapid freshwater adaptation as we observed in Scout Lake.

## Discussion

### Jackpot carriers mediate rapid freshwater adaptation

We studied the rapid adaptation of an anadromous Threespine Stickleback population to a new freshwater environment by comparing genomic variation during the first few generations after the population was founded and in contrast to the population from which it was derived[21]. Adaptation appears to have been driven primarily by a few individuals with large haploblocks of freshwater adaptive alleles (i.e., jackpot carriers) that produced families that progressively dominated

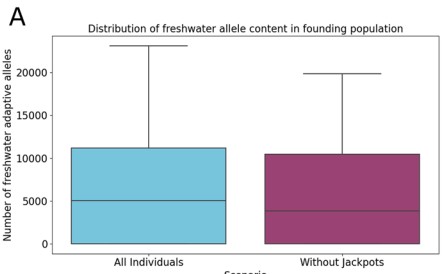

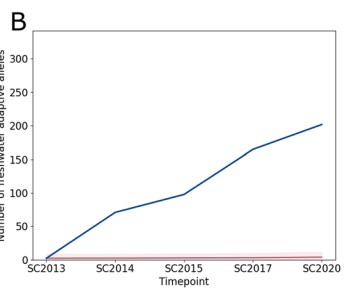

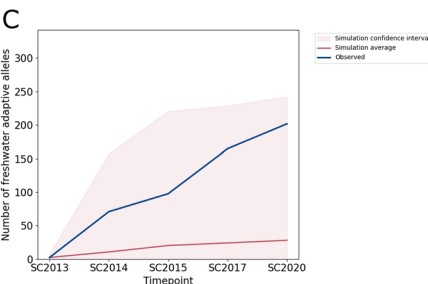

**Fig. 8 | Testing jackpot-mediated rapid adaptation with forward-in-time simulations. A** Distribution of total freshwater adaptive alleles with and without jackpot carriers in the founding population. We calculated the total freshwater adaptive alleles by summing all the alleles in all individuals used to found the population (3000); $n = 100$ replicates. Proportion of freshwater adaptive alleles at sampled comparable generations when there are no jackpot carriers in the founding population (**B**) and when jackpot carriers are present in the founding population (**C**). In (**B** and **C**), the shaded region represents the range (minimum and maximum values) of the data, while the solid lines is the mean.

in the population during the first nine years after founding. Jackpot carriers were presumably present among the anadromous Rabbit Slough founders released into Scout Lake. However, they were at such low frequencies that they were undetected in a sample from the F1 generation in the lake (i.e., SC2013). These jackpot carriers had markedly higher fitness in Scout Lake compared to non-jackpot individuals, driving the process of freshwater adaptation via population growth from just a few ancestors. This process conflicts with the transporter hypothesis[16] in that freshwater adaptation occurs via the spread of large pre-existing haploblocks of freshwater-adaptive alleles within a few generations in the lake rather than during many generations of recombination of freshwater-adaptive alleles derived from numerous founders with a few freshwater-adaptive alleles each. Jackpot carrier mediated rapid adaptation is further supported by recombination suppression in individuals that are heterozygotes at freshwater adaptive loci[19] as well as our observation of no reduction in the rate of LD decay in the first few generations in Scout Lake.

The sudden appearance of jackpot individuals in 2014 could be due to a) F2 jackpot carriers having an increased rate of survival throughout their life-cycle compared to non-jackpot individuals or b) F1 jackpot carriers having increased reproductive success. Previous experimental work has demonstrated that in freshwater families (derived from both marine and freshwater parental ancestries) which were completely plated grew slower than those families with reduced plates[38]. Although growth rate potentially impacts survival over the winter period[39], we found no size difference between jackpot carriers and non-jackpot individuals from the 2014 sample (Supplementary Note 2). On the other hand, there appears to be no difference in hatching rates between marine and freshwater families in freshwater[38], which would be expected given that anadromous fish by definition breed in freshwater. Thus, there is little evidence that would support either hypothesis of greater survivorship or greater reproductive success of the jackpot F1s. Additional research is therefore required to discriminate between these two alternative scenarios.

If the jackpot carriers observed in the SC2014 sample descended from anadromous-freshwater hybrids that we released into the lake in 2011, we expect approximately 50% of their alleles at freshwater-adaptive loci to be freshwater if they were first-generation hybrids, 25% for first-generation backcrosses to anadromous stickleback, and 12.5% for second-generation backcrosses, assuming that rare hybrids subsequently backcrossed to non-jackpot carriers with anadromous ancestry that are likely typical of Rabbit Slough fish. The content of freshwater alleles in the jackpot carriers identified in the SC2014 sample ranged from 12.1% to 49.2%, with a mean of 33%, suggesting that these individuals were first to third-generation anadromous-freshwater hybrids. The presence of recent generations of hybrids in the founding population indicates significant gene flow

from a freshwater population to the anadromous Rabbit Slough population that we used to found the Scout Lake population, and this is probably a key element of the process by which oceanic Threespine Stickleback adapt to new freshwater environments, as suggested by the transporter hypothesis[10,16]. It should be noted that we were able to rule out the possibility that recent hybrids were produced from mating between anadromous founders and very rare freshwater individuals from the original Scout Lake population that had survived the rotenone treatment of the lake (Supplementary Note 3).

Bassham et al.[17] studied multiple natural freshwater Threespine Stickleback populations that were more than 50 years old[40] and suggested that freshwater-adaptive alleles could enter new environments as colocalized haplotypes (i.e., as large haploblocks) but that multiple colonization events would be necessary to reassemble fully freshwater ecotypes. However, our results show that introducing about 3000 anadromous sticklebacks within 30 days was sufficient to include enough jackpot carriers for adaptation to proceed. The introduction of about 3000 anadromous sticklebacks from Rabbit Slough to two other Cook Inlet lakes in 2009[21] and 2019[25] also established rapidly adapting populations. While our experimental set up mimicked natural colonizations as much as possible, we note that Scout Lake represented a somewhat simplified ecological context as it was originally cleared of certain competitors through rotenone treatment prior to the founding of the Threespine Stickleback population, even though some competitors were also subsequently re-stocked. Future work utilizing alternative study designs will be essential to determine how generalizable our results are.

Assuming a frequency of jackpot carriers of 0.1% in the oceanic environment, as estimated previously[12,17], the probability of observing at least one jackpot carrier under binomial sampling among the approximately 3000 anadromous fish we transplanted into Scout Lake is 0.95. However, the probability of capturing at least 5, 7, or 10 jackpot carriers is only ~0.2, 0.03, and 0.001, respectively. Given that the jackpot individuals in the Scout Lake population in 2014 and subsequent generations in 2015, 2017, and 2020 primarily consist of a closed network of related jackpot carriers, it appears that adaptation of anadromous stickleback to freshwater can occur with approximately five descendants of recent oceanic-freshwater hybrids in the founding anadromous populations. Our forward-in-time simulations also showed that jackpot carriers were needed to replicate the rapid adaptation we observed in Scout Lake.

Non-jackpot individuals may also be required for anadromous stickleback to successfully colonize freshwater by anadromous stickleback. At such low frequencies, the probability of jackpot individuals mating with each other during the earliest years may be low, particularly when there are no noticeable phenotypic differences between jackpot and non-jackpot individuals (though behavioral differences

may increase the chance of this). Mating occurred between jackpot and non-jackpot individuals before 2014, as we found relatedness compatible with half-sibling and half-avuncular relationships in both 2013 and 2014. Mating with multiple non-jackpot individuals would provide a mechanism for jackpot haploblocks to increase rapidly in frequency in subsequent generations, especially given that Threespine Stickleback nests can contain more than 300 eggs[26,41]. Low recombination rates at freshwater-adaptive loci[12,19] will also ensure that large heterozygote haploblocks are inherited largely intact by numerous offspring during each generation. Thus, jackpot haploblocks could increase their frequency by orders of magnitude in just one generation of mating simply by jackpot carriers pairing with non-jackpot individuals.

### The consequences of bottlenecks on genetic load during rapid adaptation

Previous studies have proposed that population bottlenecks may be important during freshwater adaptation by oceanic Threespine Stickleback[42,43]. These bottlenecks have been inferred from an observed reduction in genetic diversity in established freshwater populations[42–44]. Here, we present the first direct observation of the temporal dynamics of stickleback adaptation to freshwater during the first nine years of colonization. We observed a bottleneck that occurred during the third and fourth year after founding, likely as a result of reduced fitness of non-jackpot individuals that made up the majority of the founding population. The resulting reduction in population size coincided with a major shift in the genetic composition of the individuals that remained in the lake, indicating the population decline was non-random and driven by strong positive selection for a small number of jackpot carriers.

However, population growth from a few related individuals will decrease genetic variation and potentially increase the population's genetic load. Genetic load can either be expressed phenotypically or masked[33] in diploid populations, while bottlenecks can increase phenotypic expression of a population's genetic load[32–34] by exposing rare, recessive deleterious alleles normally masked in heterozygotes. In addition, inbreeding after the bottleneck could increase the frequency of rare deleterious alleles and their probability of homozygosity[32,33,36]. Such a scenario can lead to one of two outcomes: a) the population could be at risk of extinction due to the increase in the fixation probability of recessive deleterious mutations[45–47] through drift, or b) negative selection could be more effective at eliminating slightly deleterious mutations, as selection is more potent against recessive or partially recessive mutants in the homozygous state[48].

In Scout Lake, the latter outcome could, therefore, increase the population's overall adaptability by purging deleterious variants that circulated at low frequencies in the anadromous ancestor alongside the positive selection of freshwater-adaptive alleles, at least during the early stages of adaptation, before new deleterious mutations could emerge. Such a process has been observed in many other species undergoing extreme bottlenecks[49,50]. Although inbreeding may have led to a general purging of deleterious mutations genome-wide, it is interesting to contrast its effects at adaptive loci with regard to the dynamics of freshwater haploblocks. While much of the genome is becoming increasingly homozygous during the earliest stages of freshwater adaptation, adaptive loci are becoming more genetically diverse as homozygous oceanic haploblocks are transitioning into heterozygous haploblocks as the freshwater alleles are increasingly positively selected for. This may allow deleterious alleles within or closely linked to freshwater haplotypes to hitchhike to higher frequencies, thus counteracting the decrease in load elsewhere in the genome. The low recombination rates keeping such haploblocks intact over long periods may act somewhat like inversions, which accumulate genetic load in other species, such as *Heliconius*[51]. Low recombination rates may even slow the speed of fixation of freshwater-adaptive alleles

later in the adaptive process, as these deleterious alleles become homozygous, again analogous to frequency-dependent selection observed in *Heliconius*, though unlike inversions, the presence of recombination hotspots between adaptive loci may allow rescue from high genetic load. Indeed, even in established populations, we rarely see individuals that are fully homozygous for all freshwater-adaptive alleles, and emerging empirical studies are exploring the dynamics of deleterious alleles linked to adaptive loci in low recombination regions (such as in Threespine Sticklebacks)[52,53].

In summary, recombination is a fundamental part of the Schluter and Conte transporter model[16] to describe parallel adaptation by oceanic ecotypes to freshwater environments. It has been clear for a long time that most alleles that encode the differences between freshwater residents and oceanic stickleback exist as standing genetic variants in the oceanic populations[6,10,13,25,54]. In cases of very rapid adaptation, the waiting time for mutations to generate new alleles[55] should take several thousand years in lake stickleback populations, and even recombination acting to combine multiple, ancient freshwater-adaptive alleles within individual genomes may take too long. The population would have to rely on nearly intact multiple adaptive alleles that entered the population by constant gene flow between the oceanic population and genetically divergent freshwater populations. Using whole genome resequencing of samples from a time series of a rapidly adapting population, we showed that rare individuals with larger haploblocks of adaptive variants maintained for at least a few generations via reduced recombination influenced whether a new population can adapt quickly to a new environment. Additional studies are required to determine if the adaptive divergence that we observed is characteristic of freshwater colonization by Threespine Stickleback populations and other species. Future research could examine whether F2 jackpot carriers have higher fitness due to their higher survival in freshwater environments or due to the reproductive success of their F1 parents.

## Methods

### Sample collection, library preparation, and genome sequencing

All fish used in this study were collected according to an approved protocol from the Institutional Animal Care and Use Committee (IACUC 1446584) at Stony Brook University and from The College of New Jersey Institutional Care and Use of Animals Committee (protocols 1908-001MW1A1 and 2002-001MW1A3). Samples of Threespine Stickleback have been collected from Scout Lake at least once a year around the start of the breeding season in late May or June since 2012. Minnow traps with a mesh of 6.35 mm or usually 3.175 mm have been set yearly for up to 24 h at less than 2 m depth and 5 m from shore. After capture, sticklebacks are separated from other fish species and transferred to a bucket of lake water to which equal volumes of sodium bicarbonate and MS-222 (tricaine methane-sulfonate) have been added at a high enough concentration to cause the fish to lose equilibrium within 30 s and to die within a few minutes. Death was inferred from the failure of the fish to react to tapping the side of the bucket in which they were held and then to pinching the caudal fins of selected fish. We netted the fish out of the bucket, washed them in lake water, and dropped them into 70 % ethanol in deionized water in a 1-liter bottle with up to about half fish and the remainder with ethanol solution. The ethanol was replaced with a fresh 70% ethanol solution within about 24 h after the lipid from the fish had discolored the ethanol with a yellow hue. The right pectoral fin was usually clipped from the specimen and placed into a small, numbered conical tube of 70% ethanol for DNA extraction, and the remainder of the fish was placed with another fish from the same sample in a numbered, 15 ml tube with 70% ethanol and one fish head up and the other head down. These fish are stored either in David M. Kingsley's laboratory in the Department of Developmental Biology, Stanford School of Medicine or in the Veeramah lab at Stony Brook University.

DNA was extracted using the Qiagen DNeasy 96 Blood & Tissue Kit for animal tissue and was quantified with a Qubit Fluorometer 3.0 set to High Sensitivity option for dsDNA. For the high-coverage DNA sequences from SC2020, we selected high quality samples with minimal evidence of fragmentation to sequence. The average concentration of the 20 selected samples was 72.32 ng/uL, ranging from 41.2 ng/uL to 134.8 ng/uL. We sent the DNA for resequencing at the Beijing Genomics Institute (BGI) using their proprietary DNBseq technology. For the low-coverage genomes, we randomly selected DNA from 96 specimens with high concentrations of DNA for each time point (except SC2014, in which there were only 48 specimens) for library preparation with plexWell™ 384 (SeqWell, Beverly, MA, USA). The plexWell™ 384 kit is based on seqWell's proprietary TN5 transposase that involves the insertion of Illumina i7 adapters into individual DNA samples (up to 96 samples), which are then pooled together. An Illumina i5 adapter is inserted into each pool and amplified in a PCR reaction using manufacturer-provided library primer mix and Roche's KAPA HiFi DNA Polymerase readymix. The concentration of each pool was within the manufacturer-recommended range of 4-8 ng/μL (RS2019-4.2 ng/μL, SC2013-5.68 ng/μL, SC2014-6.84 ng/μL, SC2015-4.56 ng/μL, SC2017-5.24 ng/μL, Crosses-7.56 ng/μL). We performed further quality controls by running aliquots of the prepared libraries on 2% agarose gel and an Agilent 2100 Bioanalyzer. We pooled all individuals on a 96-well plate together and added a unique i5-barcoded adapter. Each population of pooled 96 individuals is then sequenced on an Illumina HiseqXten platform, with paired ends read length of 150 bps at an output of around 45 Gb. Threespine Stickleback has a genome size 460 Mb, so 45 Gb output yields approximately 100X coverage.

## Crosses from Mile 87 lake for validation of called genotypes at freshwater adaptive loci

In early June of 2021, adult sticklebacks were collected from Mile 87 Lake (60.914517°, −149.101545°) using unbaited, 10 mm mesh minnow traps set within a few meters of the shoreline. Anadromous and freshwater males and females in breeding conditions were transported live in aerated lake water to the temporary laboratory facility in nearby Anchorage, Alaska. We identified ecotype identity based on substantial size and body shape differences between the adults and the anadromous (larger) and freshwater (smaller). We produced four crosses via in vitro fertilization: male anadromous x female anadromous, male freshwater x female freshwater, male anadromous x female freshwater, and male freshwater x female anadromous. We euthanized males in an overdose of buffered MS-222, and their testes were dissected and macerated in a drop of embryo medium (2ppt Instant Ocean Sea Salt in distilled water + 1 drop of methylene blue/2 L). We stripped ovulated eggs from a gravid female into a Petri dish and subsequently covered them with the extracted sperm. After three minutes to allow fertilization, we rinsed clutches thoroughly, submerged them in fresh embryo medium, and stored them at 7 °C. The following day, we separated embryos, removed dead/unfertilized individuals, and refreshed the embryo medium. We maintained clutches each day. Prior to five days post fertilization, clutches were shipped overnight to The College of New Jersey in Ewing, NJ. Embryos began hatching ~12–14 days pf, and fry were reared for an additional three weeks when they were euthanized in an overdose of buffered MS-222 and preserved in 100% ethanol for subsequent DNA extraction. Fish were collected under Alaska Department of Fish and Game permit P-21-006.

## Sexual maturity scoring of Scout 2012 and 2013 samples

We removed the gonads of females and males to perform macroscopic inspections of them. Ovaries of females were assigned to six reproductive phases following[56,57]: latent, early maturing, late maturing, mature, ripening, and ripe. These phase assignments have been updated with more recent terminology based upon microscopic

histological observations[58] but were used as initially designated. Latent females were considered sexually immature, whereas females in the other five phases were considered sexually mature. Testes of males were assigned to latent or mature phases based on their size and opacity. Latent males had small, transparent or translucent testes, and mature males had enlarged, cloudy to opaque white testes.

## Catch per unit effort estimation

Catch per unit effort (CPUE) is the total number of fish caught divided by "trap h" The number of trap h is the product of the number of traps and the number of h we set. We captured all Threespine Sticklebacks using 1/8 or ¼-inch mesh Gee Minnow Traps set overnight for <24 h within <5 m of shore and in water <1.5 m deep. We placed the traps in and adjacent to patches of rooted aquatic plants if they had begun to grow. The bottom was mostly sand or silt. We set about 100 traps behind the same three residences on the north shore of the lake.

## Bioinformatics processing

Sequencing reads were de-multiplexed into individual genomes based on the i7-barcodes by BGI. The total number of bases sequenced per sample ranged from 0.004 Gb to 1.79 Gb, with a mean of 0.54 Gb. The total number of reads per sample ranged from 21,266 to 11,917,120, with a mean of 3,028,575 reads. After trimming adapters with AdapterRemoval (ver. 2.2.2)[59], we mapped reads to the Threespine Stickleback genome version gasAcu1-4[12,60]. Libraries sequenced across multiple Illumina lanes and runs were processed separately, and we added read groups using Picard before being merged with samtools[61]. We also marked duplicates using Picardtools. We performed base recalibration using BaseRecalibrator from GATK version 3.7[62,63].

## Genotype calling and validation

As mean coverage was ~1x across samples, we developed a novel method for calling diploid genotype states at freshwater-adaptive loci. As described in the Results, we identified highly linked SNPs within loci that underlie freshwater adaptation, using the correlation coefficient (≥0.99) of SNP allele frequencies from Pool-Seq experiments covering a time series of lake populations undergoing freshwater adaptation: Loberg, Cheney, and Scout Lakes[12] (Supplementary Data 5). These multi-SNP haplotypes allowed us to call the diploid state of each locus based on an approximate genotype likelihood approach that employed multi-SNP haplotype information. Genotype likelihoods were calculated for individual SNPs within each locus using the approach described in DePristo et al.[64] and then summed across correlated SNPs based on the freshwater or anadromous allele to obtain approximate likelihoods for anadromous homozygous, heterozygous, and freshwater homozygous allelic states for each locus. Genotype calls were assigned for a locus if there was a minimum of 3 SNPs with reads with mapping error probabilities less than 0.001 and base call error probabilities less than 0.01. We coded freshwater-adaptive loci that did not meet these filtering criteria as missing.

We validated the ability of this method to determine the diploid state of the loci by applying it to call haplotypes of a set of experimental crosses between anadromous and freshwater parents from the impoundment at mile 87 along Seward Highway. We observed that estimated states from our likelihood method were highly consistent with expectations from the results of our crosses (Fig. S1). Parents were scored as homozygous for the alleles associated at most loci with their ecotype. More importantly, offspring in these experimental crosses inherited states consistent with the states of their two parents. We did not consult that parental genotypes were to call genotypes of their progeny. For example, the offspring of parents with alternative homozygous states at the locus were invariably scored as heterozygous.

We also imputed our low-coverage data with Beagle 4.0[65] to enhance genotyping accuracy using a reference panel of 169 medium-

to high-coverage whole genomes (see below). We then adjusted our likelihood approach to incorporate imputation-based genotype probabilities rather than individual SNP genotype-likelihoods. The overall pattern of the genotypic state of each locus was not qualitatively different between our likelihood approach and Beagle-imputed genotype probabilities (Fig. S1), as the dosage of freshwater-adaptive alleles carried by each individual was similar in both methods. Beagle imputation led to a reduced rate of missingness per locus (280 loci without missingness after vs. 235 before imputation with Beagle).

## Imputation of low-coverage data with Beagle 4.0

Using Beagle 4.0 and a reference panel of 169 genomes, we imputed the missing data in the low-coverage sequences. The reference panel comprised genomes mapped to the Gas-Acu1-4 genome, with bwa-mem and base recalibration performed on the bams with GATK's BaseRecalibrator using hard-filtered SNPs from Roberts-Kingman et al.[12]. The coverage of the included genomes ranged from 9–62X (Supplementary Data 3). It included 69 oceanic fish, 58 from well-established freshwater populations, 40 from experimental transplants, and two of unknown ecotype designation with a global distribution from refs. [12,66]. We followed GATK's best practices guidelines for variant calling with HaplotypeCaller, joint genotyping, and variant quality score recalibration with VariantRecalibrator. We only included biallelic sites in the reference panel and targeted 6,027,376 high-quality SNPs. We downloaded Beagle 4.0 from https://faculty.washington.edu/browning/beagle/beagle.r1399.jar to perform haplotype phasing per chromosome and imputation on all low-coverage data simultaneously. The genotype likelihood used was estimated using GATK's HaplotypeCaller. We used recombination maps from ancestral Rabbit Slough, which was previously estimated[12] using LDhelmet[67]. We filtered the recombination maps to exclude absent sites in the reference panel. Then we ran Beagle 4.0 as follows java -Xmx90G -jar beagle.27Jan18.7e1.jar gl=chrII.gz ref=WGS_170_recalibrated_Snps6mil_chrII.phased.vcf.gz map=chrII.map impute=True out=All_samples_chrII. After imputation, we combined all chromosomes using bcftools concat.

## Haploblock construction and erosion

To assess the impacts of imposing a gap threshold on the observed distribution of defined haploblocks for the freshwater adaptive loci (Fig. 3, Supplementary Data 2), we analysed the haploblock distributions while imposing a physical distance threshold. We first filtered sites with missing data (as was the case for our initial analyses presented in the main manuscript). We chose the maximum gap threshold as twice the size of the largest adaptive locus, which was -0.4 Mb (excluding the three well-known inversions on chrI, XI and XXI, which are significantly larger than most adaptive loci and span up to 1.9 Mb). Thus, if adjacent loci with freshwater alleles are more than 0.8 Mb apart, we break any single larger haploblock into two smaller haploblocks. When we compared the haploblock distributions when imposing a threshold of 0.8 Mb (Figs. S4, S5) with those estimated in the original analysis, we do see smaller haploblocks in both physical and genetic distances, which is to be expected as blocks are broken up. However, more importantly, when imposing this 0.8 Mb threshold we still see no major increase in haploblock size across time points using both physical and genetic distance compared to the rapid increase in per individual freshwater content, consistent with our original inference from Fig. 3, and there is a very strong correlation in mean haploblock size with and without imposing the 0.8 Mb threshold (Fig. S18).

## Classification of individuals as jackpot carriers

We performed a dip test to check the modality of the SC2014 sample based on freshwater content (i.e. number of freshwater alleles per individual) and identified two modes. We also pooled all samples across timepoints for the same metric of the number of freshwater alleles per individual, and used a Bayesian Gaussian Mixture Modelling (BGMM) approach implemented in scikit-learn (version 1.7.1) in Python 3.11.13 to estimate the number of underlying distributions that best fit the data. BGMM incorporates a Dirichlet process prior that allows the model to infer the effective number of components directly from the data. We set the maximum number of components to four and used the full covariance type, which treats each component to have its own general covariance matrix. However, doubling the number of maximum components did not change the result of the analysis. Bayesian information criterion (BIC) suggested two components to be the best choice, consistent with the dip test (Fig. S6). The fitted model provided posterior probabilities for each observed individual's membership in each of the two main components based on their freshwater content. We then categorized all the datapoints at each timepoint into one of these two components (Fig. 2). The model categorized individuals with 5% freshwater adaptive alleles as jackpot carriers with a posterior probability of 0.95. Individuals with 8% freshwater adaptive alleles were classified as jackpot carriers with 100% certainty (Fig. S7).

## Estimating the site frequency spectrum (SFS) with theta and Tajima's D

To account for our low-coverage genome data, we used RealSFS from ANGSD[68] to estimate each sample's site frequency spectrum (SFS) based on genotype likelihoods. For SC2015, we identified four duplicates through our relatedness analyses, which were removed before estimating the SFS. We masked ChrXIX and all transposons before estimating the SFS. We used the ancestral genome constructed by Roberts Kingman et al.[12]. Then we imported the estimated SFS into dadi[69] and folded the SFS, and performed further analyses, including summary statistics like nucleotide diversity ($\pi$), Watterson's estimator of $\Theta$ and Tajima's D[70]. To prevent any bias introduced by unequal sample sizes of various time points, we used dadi to project down each time point to 40 samples. For 0-fold and 4-fold degenerate sites, we used degenotate (https://github.com/harvardinformatics/degenotate) to compute the degeneracy of coding sites across the stickleback genome. In total, we identified 24,891,530 0-fold and 6,348,969 4-fold degenerate sites. Then, we used the -sites option from ANGSD to estimate the SFS for the 0-fold and 4-fold sites.

## Linkage disequilibrium (LD) decay

To estimate the LD decay across the sampling period, we focused on the 280 adaptive loci without any missing data. For each time point, we estimated pairwise $r^2$ among all combinations of the 280 loci using hard-called locus-based genotypes, coded as 0 for homozygous freshwater, 1 for heterozygous, and 2 for homozygous marine, corresponding to the number of marine alleles carried by each individual at each locus. To quantify the uncertainty in LD estimates, we performed bootstrapping using genotype matrices for each of the timepoints. For each timepoint, the genotype matrix was resampled with replacement 1000 times to generate a distribution of $r^2$ values in order to estimate 95% confidence intervals. We then binned pairs of loci/SNPs by distance to examine differences in the rate of LD decay.

## Kinship analysis

We utilized two methods to infer pairwise relatedness, both designed for low-coverage data, READv2[29] and NGSrelate[30]. We called genotype likelihoods genome-wide for all SNPs identified previously by Roberts Kingman et al.[12] using GATK Unified Genotyper v3.7.0[64]. We estimated allele frequencies for each Scout Lake sample separately using the approach of Kim et al.[71]. We then intersected these allele frequencies with allele frequencies estimated from pool-seq data from Roberts Kingman et al.[12] from the same time points using the approach of Lynch et al.[72]. For each time point, any SNP that had a total depth of coverage (across all samples) of less than half or double the mean coverage for both the TN5 and pool-seq data was included in the

genome mask. In addition, we performed Fisher's exact test on a 2 × 2 contingency table containing the depth of each allele in the TN5 and Pool-Seq data and added any SNPs with a *p*-value less than 0.05 to the mask. We merged masks from each time point, known repeats, and transposable elements. Then, we applied this mask to all time points, resulting in a set of 2,119,671 high-quality SNPs for relatedness analysis. We made haplodiploid genotypes for use in READv2 based on the highest allele depth[28,29], which estimates the degree of relatedness between pairs of individuals by using the proportion of allele mismatches across the genome. We used READv2[29], which estimates up to third-degree relatedness. We validated READv2 results using ngsRelate[30], which utilizes population allele frequencies to estimate identity-by-descent probability. We ran ngsRelate with options ngsRelate -F 1 -h input.vcf -O output -z samp_list -A AF, where AF is the allele frequency estimated from a set of unrelated samples from SC2013 (again estimated using the approach of Kim et al.[71])

We used Cotterman coefficients[73] for first-degree relatives to distinguish parent-offspring from sibling pairs. We relied on the Cotterman coefficients estimated from ngsRelate instead of READv2. When there is no inbreeding, the Jacquard coefficients J9, J8, and J7 map to the Cotterman coefficients. Based on the Cotterman coefficients, the five first-degree relatives observed in our Scout Lake samples were likely siblings. However, READ2 designates 4 out of these five as parent and offspring. In READv2, the Cotterman coefficients k0 and k2 correspond to the proportions of windows classified as unrelated and identical. This proportion is expected to be low for parent-offspring pairs and higher for siblings.

### Inbreeding estimation

We used NgsRelate[30] to estimate individual inbreeding coefficients using allele frequencies from unrelated specimens from SC2013. To validate the estimates of inbreeding coefficients from ngsRelate (Fig. 6D), which is designed for low-coverage datasets, we used PLINK to estimate the runs of homozygosity for our high-coverage SC2020 samples. We compared the results with estimates of inbreeding from ngsRelate for them. We estimated runs of homozygosity by first performing LD pruning in PLINK with options --indep-pairwise 50 5 0.5. We then estimated the runs of homozygosity with PLINK, option --homozyg --homozyg-window-kb 5000. There is a high correlation between the runs of homozygosity and estimated inbreeding coefficients from ngsRelate ($r = 0.86$, Fig. S19).

We also downsampled the high-coverage SC2020 samples to 1X coverage using samtools (v1.9) with option samtools view -s x -b input.bam -o downsampled.bam, where x is calculated as 1/y, and y is the coverage of the high-coverage sample estimated using GATK's DepthOfCoverage. We used ngsRelate to estimate the inbreeding coefficient using allele frequencies from SC2013. There is a high correlation between estimates from downsampled and high-coverage samples ($r = 0.822$, Fig. S11).

### Forward Wright-Fisher simulation

We performed forward-in-time Wright Fisher simulations to model the process of adaptation with and without jackpot individuals using SLiM v5[37,74]. We assumed non-overlapping generations, with a generation time of one year, following the framework of refs. 12 and 18. We simulated a large oceanic (anadromous) population connected to 10 distinct freshwater populations (Fig. S16). We allow migration to occur between the freshwater populations and the marine population, but not among freshwater populations. As in Roberts Kingman et al.(2021), we set the migration rate from marine to freshwater populations (M_AN_TO_FW) to be 0.001, and migration rate from freshwater populations to marine population (M_FW_TO_AN) to be 0.01. At the start of the simulation, we introduce freshwater-adaptive mutations at all 341 loci in one of the freshwater populations to mimic an established population that is fully adapted to a freshwater environment.

We do not allow de novo mutations during the simulation and each mutation has a selection coefficient of 0.01 in the freshwater environment. The value of 0.01 is of similar magnitude to the mean selection coefficient inferred across loci in Roberts Kingman et al.(2021). At generation 1000, we established a new freshwater population representing Scout Lake by randomly sampling individuals from the anadromous population. Following its founding, no further migration was permitted between this simulated Scout Lake and any other populations in the system.

### Phenotypic characterization

We measured three traits that differ consistently between anadromous and freshwater stickleback[75] and have diverged rapidly from the ancestral anadromous condition in other recently found lake populations in the SC2014 sample. The three traits were lateral plate morphs, gill rakers and standard length (Supplementary Note 2).

### Statistics and reproducibility

We performed statistical analyses in Python. Sample sizes are reported in the figure legends. All data and code necessary for reproducing the conclusions in this study have been made publicly available.

### Reporting summary

Further information on research design is available in the Nature Portfolio Reporting Summary linked to this article.

## Data availability

The whole genome data generated in this study have been deposited in the Sequence Read Archive (www.ncbi.nlm.nih.gov/sra) under accession code PRJNA1231081. Whole genomes from Rabbit Slough previously published in ref. 12 can be found at SRA under accession code PRJNA671690. All other whole genomes from ref. 12 can be found under accession code PRJNA247503. The data to generate the figures can be found in the Code Ocean capsule https://codeocean.com/capsule/9257557/tree (https://doi.org/10.24433/CO.2139373.v1).

## Code availability

All custom scripts have been deposited to GitHub and can be accessed through the following link: https://github.com/a-kwakye/Rare-Jackpot-Individuals-Drive-Rapid-Adaptation-in-Threespine-Stickleback[76] and Code Ocean at https://codeocean.com/capsule/9257557/tree (https://doi.org/10.24433/CO.2139373.v1).

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

## Acknowledgements
This work was supported by NIH R01GM124330 to K.R.V. and M.A.B., NSF grants DEB-0322818 and DEB-0919184 to M.A.B. and F. J. Rohlf, and the Newcomb College Institute of Tulane University to D.C.H. We would like to thank the many students who helped found the Scout Lake population and to capture stickleback from it. In particular, we thank W. E. Aguirre and P. J. Park, who were instrumental in the fieldwork. We thank members of the Veeramah Lab for comments and discussions during the preparation of this manuscript, especially Thomas Bertino and Rachael Herman for useful discussions on simulations.

## Author contributions
A.K: Formal analysis, DNA quantification, library preparation, genomic data processing; K.R: experimental design, DNA extraction, DNA quantification, library preparation, construction of reference panel for imputation, genomic data processing; M.W: crosses from Mile Lake; D.C.H: sampling, life-history data collection and processing; M.A.B: experimental design, population founding, most sampling, tissue sampling, and morphological data; K.R.V: experimental design, project supervision. A.K wrote the initial draft with input from all authors. All authors contributed to production of the final version.

## Competing interests
The authors declare no competing interests.
