## [Transparent Peer Review file · Nature Communications]

Rare Jackpot Individuals Drive Rapid Adaptation in Threespine Stickleback

Corresponding Author: Dr Krishna Veeramah

Version 0:

Reviewer comments:

Reviewer #1

(Remarks to the Author)

The manuscript by Kwakye et al. addresses an intriguing question about the genomic mechanisms underlying rapid adaptation to freshwater environments in Threespine Stickleback, a model organism for evolutionary biology. This work challenges the traditional recombination-centered "transporter hypothesis" by demonstrating empirically that rapid freshwater adaptation can be mediated primarily through rare individuals carrying large haploblocks of adaptive alleles (jackpot carriers), rather than gradual recombination events. The study leverages an exceptionally rich temporal genomic dataset and offers insights into kinship dynamics, population bottlenecks, genetic load management, and the initial absence of phenotypic divergence, enhancing our understanding of evolutionary mechanisms. Nevertheless, several points should be addressed to strengthen the manuscript further:

1. The authors used a 2019 Rabbit Slough sample as the ancestral reference for detecting standing genetic variation underlying rapid adaptation. However, the actual introduction event occurred in 2011, making a 2009 or 2011 sample more appropriate as the baseline. The choice of a 2019 baseline may introduce biases from genetic drift or selection occurring over the intervening period. The authors should clearly justify using the 2019 sample or preferably reanalyze key results with the available 2009 data to ensure robust identification of adaptive loci.
2. The authors mention that the SC2013 samples may include a mixture of F0 founders and early-born F1 individuals. However, the manuscript does not explain why no genetic substructure or differentiation between these two groups is observed within the SC2013 cohort. This point should be explicitly addressed.
3. The authors define jackpot carriers as individuals with more than 10% freshwater-adaptive alleles, based on the bimodal distribution observed in the SC2014 sample. While this threshold is reasonable, it would strengthen the methodological rigor if the authors could briefly discuss whether alternative thresholds (e.g., 8% or 12%) were tested and how sensitive their key results (e.g., haploblock expansion patterns, relatedness networks) are to the exact cutoff chosen. Additionally, it would be informative to comment on whether the definition of jackpot carriers remains consistent across later timepoints (e.g., SC2015, SC2017) as selection proceeds and freshwater dosage distributions shift.
4. While the manuscript defines a contiguous haploblock as consisting of two or more consecutive freshwater-adaptive loci, it remains unclear whether a maximum physical distance threshold was imposed between adjacent loci, or how missing data across loci were handled during block construction.
5. Given the progressive increase in inbreeding coefficients across timepoints, it would be valuable for the authors to briefly discuss whether increased homozygosity could have influenced the observed patterns of haploblock expansion or adaptive allele frequency changes.
6. As recombination could theoretically break apart large freshwater haploblocks over successive generations, it would be useful if the authors could briefly discuss whether any evidence of haploblock erosion (e.g., internal recombination events, partial block loss) was observed by SC2017 or SC2020.
7. While the authors clearly state that Scout Lake was cleared of predators and competitors through rotenone treatment before stickleback introduction, they do not discuss how this simplified ecological context might have influenced the rapid adaptation process observed. The absence of ecological interactions such as predation and interspecific competition could fundamentally alter selective pressures, potentially making Scout Lake a special case. I recommend that the authors explicitly address this point in the Discussion and temper the generalization of their findings accordingly.

Reviewer #2

(Remarks to the Author)

This study by Kwakye and colleagues investigates the genomic mechanisms of rapid freshwater adaptation in Threespine Stickleback. Using time-series genomic data, they propose that adaptation was primarily driven by rare "jackpot" individuals present in the founders, who carried large, pre-existing haploblocks of freshwater-adaptive alleles. These individuals and their descendants are suggested to have rapidly increased in frequency following a population bottleneck, leading to inbreeding and potential purging of deleterious alleles. This proposed mechanism offers a valuable contrast to the traditional transporter hypothesis.

Major Suggestions:

1- The central claim regarding the indispensability of jackpot carriers for the observed rapid sweep would be substantially strengthened by quantitative modelling. Employing explicit ABC or forward simulations could test this hypothesis against alternatives and help disentangle selection from genetic drift, particularly during the bottleneck.

2- the argument for early physiological adaptation preceding morphological changes, currently based largely on negative evidence, would be more compelling with direct positive evidence. Measuring relevant physiological traits, such as osmoregulation or metabolic rates, in early-generation individuals could provide this. Or tune down it or move to the supplementary to save text in the main text (see below).

3- the manuscript's length and density could be reduced for clarity and impact. Now it is too long for the journal. Streamlining the narrative to focus on the core "jackpot carrier" hypothesis and condensing or moving less central results, such as the detailed morphology and fads2 findings, to supplementary information is recommended.

Minor Suggestions:

1- To improve accessibility for a broader audience, the authors should enhance the in-text explanation of Fig. 6, more explicitly guiding readers through the interpretation of SFS patterns and their connection to conclusions about genetic load and selection.

2- all custom scripts and bioinformatic pipelines used for the genomic analyses should be made available.

3- The figure legend for Fig. 1F: the 344 freshwater-adaptive loci, should briefly explained.

4- The Y-axis labels for the two columns in Fig. 2 need to be clarified.

5- Fig 4 panel labels should be annotated

Version 1:

Reviewer comments:

Reviewer #1

(Remarks to the Author)

The authors have made a commendable effort, including adding new analyses (e.g., sensitivity testing of haploblock thresholds and alternate definitions of "jackpot carriers"), and expanding the discussion in several key areas. These additions improve the clarity and scope of the manuscript. However, several methodological concerns remain unresolved, and important interpretive and theoretical issues continue to undermine the strength of the main conclusions. In particular, the current analytical framework still relies on arbitrarily defined thresholds without sufficient theoretical justification or statistical validation. Moreover, potential confounding factors, such as inbreeding-driven homozygosity and recombination-driven haploblock erosion, are not thoroughly examined or incorporated into the interpretation of adaptive dynamics.

Major concerns:

1. The authors argue that the RS2019 sample was not used to identify adaptive loci and that allele frequencies are highly correlated between RS2009 and RS2019. However, this response does not fully address the original concern. The key issue is not the origin of adaptive loci per se, but whether the RS2019 sample can reliably serve as a reference for standing variation prior to the 2011 introduction event. The authors demonstrate that adaptive alleles underwent selection between 2011 and 2015, raising the possibility that some allele frequency changes also occurred in the RS population over the same period. Given this context, using RS2019 to represent ancestral allele frequencies without any temporal correction or bootstrapping introduces potential biases, even if correlations appear high. More importantly, the RS2009 sample is explicitly available (as acknowledged), and its allele frequencies were used in previous publications. There is no compelling reason not to replicate key frequency-based visualizations (e.g., Fig. 1F) using RS2009 data, or at least provide results based on both datasets to confirm robustness. Without this, the choice of RS2019 remains methodologically suboptimal and weakens the interpretation of standing genetic variation.

2. The classification of "jackpot carriers" using a fixed 10% threshold for freshwater-adaptive alleles is central to multiple downstream analyses, yet the rationale for this cutoff remains insufficiently justified. While the authors briefly explored sensitivity to alternative thresholds (8% and 12%), these appear arbitrarily selected and do not constitute a systematic evaluation. Moreover, Figure 2C shows that the freshwater content distribution shifts substantially across years, suggesting that a fixed threshold may capture biologically distinct subsets of individuals at different timepoints. This raises concerns about the comparability and interpretation of "jackpot" status across temporal scales. The authors are encouraged to provide a stronger conceptual and statistical justification for the 10% threshold, and to consider whether a relative or quantile-based definition (e.g., top x%) might yield more robust or interpretable results given the temporal dynamics of freshwater allele accumulation.

3. The authors acknowledge that elevated homozygosity resulting from inbreeding may have facilitated the purging of maladaptive alleles, thus potentially accelerating positive selection at freshwater-adaptive loci. However, this response does not fully address the potential mechanistic implications of increased homozygosity on haploblock structure and allele frequency dynamics. For instance, greater homozygosity could reduce effective recombination, leading to the apparent expansion or preservation of extended haplotypes over time. Additionally, in small populations, hitchhiking effects may be more pronounced, allowing non-adaptive alleles to rise in frequency due to linkage with adaptive loci. These possibilities warrant at least brief acknowledgment in the discussion to avoid over-simplification of the observed genomic patterns as purely selection-driven. A more nuanced consideration of demographic and genetic drift effects would strengthen the interpretation of haploblock dynamics over time.

4. While the authors reasonably argue that increasing homozygosity could obscure recombination signals, the potential erosion of large freshwater haploblocks remains an important consideration. Even in the absence of parent-offspring trios, population-level patterns, such as reduced haploblock lengths, internal heterogeneity, or linkage disequilibrium decay, could provide indirect but informative evidence of recombination. Currently, the manuscript presents no empirical analysis to support the conclusion that recombination played a limited role. Given that recombination is a central process influencing haplotype integrity and adaptive allele assembly, a lack of data-driven support weakens the strength of this claim. A brief analysis or at least an expanded discussion of potential haploblock erosion would strengthen the interpretation that haploblock persistence reflects biological processes (e.g., selection, demographic dynamics) rather than methodological limitations or overlooked recombination events.

Reviewer #2

(Remarks to the Author)

I am satisfied with the authors' thorough and thoughtful revisions. The incorporation of forward simulations, restructuring of the manuscript for clarity, and relocation of sections to the supplementary material have significantly strengthened the paper. The authors have addressed all of my concerns.

Congratulations to the authors on an excellent study.

Version 2:

Reviewer comments:

Reviewer #1

(Remarks to the Author)

The authors have fully addressed my concerns by adding multiple robust new analyses and substantially strengthening the methodological and mechanistic interpretation. The revisions are thorough and well executed, and I have no further comments.

made.

RESPONSE TO REVIEWER COMMENTS FOR:

RARE JACKPOT INDIVIDUALS DRIVE RAPID ADAPTATION IN THREESPINE STICKLEBACK” (NCOMMS-25-20702-T)

Our responses to the reviewer’s comments are below and any text added to the main text are marked red.

Reviewer #1 (Remarks to the Author):

The manuscript by Kwakye et al. addresses an intriguing question about the genomic mechanisms underlying rapid adaptation to freshwater environments in Threespine Stickleback, a model organism for evolutionary biology. This work challenges the traditional recombination-centered "transporter hypothesis" by demonstrating empirically that rapid freshwater adaptation can be mediated primarily through rare individuals carrying large haploblocks of adaptive alleles (jackpot carriers), rather than gradual recombination events. The study leverages an exceptionally rich temporal genomic dataset and offers insights into kinship dynamics, population bottlenecks, genetic load management, and the initial absence of phenotypic divergence, enhancing our understanding of evolutionary mechanisms. Nevertheless, several points should be addressed to strengthen the manuscript further:

1. The authors used a 2019 Rabbit Slough sample as the ancestral reference for detecting standing genetic variation underlying rapid adaptation. However, the actual introduction event occurred in 2011, making a 2009 or 2011 sample more appropriate as the baseline. The choice of a 2019 baseline may introduce biases from genetic drift or selection occurring over the intervening period. The authors should clearly justify using the 2019 sample or preferably reanalyze key results with the available 2009 data to ensure robust identification of adaptive loci.

We tested the correlation between the minor allele frequencies estimated from two Rabbit Slough samples (a sample of 100 individuals from 2009 sequenced using pool-seq approach in Roberts Kingman et. al¹ and the 96 low coverage genomes from 2019 used in this study). We found a very high significant correlation between the two timepoints when considering all SNPs (Review figure 1) and when we only look at SNPs within regions of the genome with freshwater adaptive alleles¹ (Review figure 2). This demonstrates that drift has not had a strong effect in the intervening years, which is expected given that the Rabbit Slough population is directly connected to the sea and hence has a large effective population size. We also note that the 2019 Rabbit Slough sample was not used to identify adaptive loci. All adaptive loci we used were identified by Kingman et al. ¹, which utilized the RS2009 sample along with pooled samples from Loberg, Cheney and Scout, including the Scout time points reanalyzed using whole genomes in this study. The RS2019 sample was simply used to demonstrate the typical

distribution of freshwater alleles in a large anadromous population sample (figure 1F), and a similar distribution is observed in the smaller sample of 2009 whole genomes and pools from Kingman et al (Review figure 3)

We have now added the sentence below to the manuscript to explicitly state that the two samples are very similar [lines 339-342], and added Review figures 1 and 2 below to the supplementary figures (fig. S3 and S4).

“The allele frequencies at freshwater adaptive loci, estimated from a sample collected in 2009 before the founding of the lake and the RS2019 sample, were highly correlated (fig. S3, S4), consistent with a large and stable anadromous population”

Review figure 1: Correlation between minor allele frequencies estimated from pool-seq of 100 individuals sampled in Rabbit Slough in 2009 and 96 individuals sampled from Rabbit Slough in 2019 used in this study, which was used in the main manuscript using all SNPs.

Review figure 2: Correlation between minor allele frequencies estimated from pool-seq of 100 individuals sampled in Rabbit Slough in 2009 and 96 individuals sampled from Rabbit Slough in 2019 used in this study, which was used in the main manuscript using SNPs within freshwater adaptive regions.

Review figure 3: Genotypes of 20 whole genomes sampled from Rabbit Slough in 2009 showing predominantly marine genotypes.

2. The authors mention that the SC2013 samples may include a mixture of F0 founders and early-born F1 individuals. However, the manuscript does not explain why no genetic substructure or differentiation between these two groups is observed within the SC2013 cohort. This point should be explicitly addressed.

We apologize if our original language was not clear about this point, but actually, we did not find evidence that supported a mixture of F0 founders and early born F1 individuals. We indicated this on lines 154-155, where we state that “*The adults stocked in Scout Lake in 2011 produced*

offspring after release but did not survive the winter (fig. 1B and Fig. 3 from Kurz et al.²).” Thus, we found through our own analyses, and that from Kurz et al² that the population in SC2013 were all F1 progeny, not a mixture of F0 founders and early-born F1 individuals (see fig. 1E). In addition, our Fig. 1 length information specifically provided context on the reproduction and cohorts in the first 4 years, where there was no evidence for a mixture of the founders and F1 in the year after founding and that the F1s only reproduced in the second year and not as one year olds. We do note that we used ‘mostly’ on line 220 in the original manuscript in the sentence beginning “*Out of the 81 individuals in SC2013 (mostly F1 generation in the lake)...*”. Perhaps this is where some of the confusion arose, where we were attempting to denote that, although our evidence does not support their presence in SC2013, it is not impossible that there might be F0 founders. Therefore we have now removed “*(mostly F1 generation in the lake)*” from that sentence. Perhaps there is another area of the manuscript we accidentally gave the impression that 2013 contained a mix of F0 and F1s that we missed, and if the reviewer could point it out, we would be happy to amend the text appropriately.

3 [part A]. The authors define jackpot carriers as individuals with more than 10% freshwater-adaptive alleles, based on the bimodal distribution observed in the SC2014 sample. While this threshold is reasonable, it would strengthen the methodological rigor if the authors could briefly discuss whether alternative thresholds (e.g., 8% or 12%) were tested and how sensitive their key results (e.g., haploblock expansion patterns, relatedness networks) are to the exact cutoff chosen.

As suggested, we adjusted the cut off to 8%, and 12% for calling jackpot carriers and reassessed our results. Changing the threshold affected the categorization of just one individual in SC2014, with a 12% cut off leading to one less jackpot carrier. In SC2015, an 8% cut-off led to the classification of three non-jackpot individuals as jackpot carriers, while increasing it to 12% excluded 4 jackpot carriers (Review figure 4). There is also one jackpot carrier in SC2017 that had fewer than 12% freshwater adaptive alleles (Review figure 4). Regardless, the pattern of growth of proportion of jackpot individuals is qualitatively the same, demonstrating that alternative thresholds have minimal effects on which specimen gets classified as a jackpot carrier, and therefore their relative distribution across the timepoints. In fact, the definition of which samples get classified as a jackpot carrier does not influence the framework of any of our analyses from a statistical perspective, except for our site frequency spectrum (SFS) analysis between jackpot carriers and non-jackpot individuals in the SC2014 sample in fig. S11, which is the only analysis where the two groups are directly compared.

Review figure 4: Distribution of jackpot carriers over time using different thresholds (8, 10, and 12%) as a definition of jackpot carrier

Of the few individuals that get re-classified based on shifting the cut off, the majority had one or two connections in our relatedness networks, except for one individual in SC2015. This individual was classified as a jackpot using a 10% cut off but did not meet the 12% cut off as a jackpot. The individual had multiple relatives, including a first-degree relative, and has been highlighted in Review figure 5. The fact that this individual is part of the larger cluster of related jackpot individuals indicates that our 10% threshold was probably the most appropriate in defining jackpots, though in general the structure of the relatedness networks is almost identical regardless of the threshold used.

Overall, shifting the threshold up or down does not lead to conclusions largely different from what we observed from the 10% we originally set in the manuscript. We have added lines 243-245 to make this point and also Review figure 5 to the supplementary figures [fig. S5] to point out the effect of altering the threshold.

Reviewer figure 5: Relatedness of samples from SC2014, SC2015 and SC2017. The highlighted individual, SC-2015-PB-X153-E11 had multiple relatives.

3 [part B]. Additionally, it would be informative to comment on whether the definition of jackpot carriers remains consistent across later timepoints (e.g., SC2015, SC2017) as selection proceeds and freshwater dosage distributions shift.

We note that our definition of jackpot carrier never changes throughout the study. It is always defined as individuals with $\geq 10\%$ freshwater allelic content. At no point does this definition change. We simply ask whether the number of individuals who meet this criteria change across the various time points.

4. While the manuscript defines a contiguous haplotype block as consisting of two or more consecutive freshwater-adaptive loci, it remains unclear whether a maximum physical distance threshold was imposed between adjacent loci, or how missing data across loci were handled during block construction.

We thank the reviewer for this important methodological observation. In the initial manuscript we did not impose a maximum threshold between adjacent loci, and we removed all loci with missing data.

We have added the following lines 163-166 to the manuscript to state this explicitly:

“We then applied this approach to the whole genomes from Rabbit Slough and Scout. Loci with missing genotype calls in any individual were excluded, resulting in 235 loci out of the original 344 that were analysed in subsequent analysis.”

And lines 183-186:

“For the results presented below, we did not impose a maximum threshold distance between adjacent loci when defining a block. However, imposing a maximum threshold did not change the overall interpretation of the haploblock distribution”.

However, we have also considered the impact of this approach and re-constructed the haploblocks by setting different maximum thresholds between adjacent loci (i.e. a gap threshold). We first filtered sites with missing data (as was the case for our initial analyses presented in the main manuscript). We chose the maximum gap threshold as twice the size of the largest adaptive locus, which was ~0.4Mb (excluding the three well-known inversions on chrI, XI and XXI, which are significantly larger than most adaptive loci and span up to 1.9Mb). Thus, if adjacent loci with freshwater alleles are more than 0.8Mb apart, we break any single larger haploblock into two smaller haploblocks. When we compared the haploblock distributions when imposing a threshold of 0.8Mb (Review figure 6 and 7) with those estimated in the original analysis, we do see smaller haploblocks in both physical and genetic distances, which is to be expected as blocks are broken up. However, more importantly, when imposing this 0.8Mb threshold we still see no major increase in haploblock size across time points using both physical and genetic distance compared to the rapid increase in per individual freshwater content, consistent with our original inference from Fig 2, and there is a very strong correlation in mean haploblock size with and without imposing the 0.8Mb threshold (Review figure 8).

Review figure 6: Comparison of physical distance for each timepoint for different cut off scenarios. A) When there is no threshold (which is reported in the main manuscript) B) When the threshold is set to 0.8Mb

Review figure 7: Comparison of genetic distance for each timepoint for different cut off scenarios. A) When there is no threshold (which is reported in the main manuscript) B) When the threshold is set to 0.8Mb

Review figure 8: Plot of average genetic distance when setting a threshold of 0.80Mb (y-axis) and when no threshold is set (x-axis).

5. Given the progressive increase in inbreeding coefficients across timepoints, it would be valuable for the authors to briefly discuss whether increased homozygosity could have influenced the observed patterns of haploblock expansion or adaptive allele frequency changes.

On lines [441-442] (which was 662-664 of the original text), we suggested that homozygosity resulting from inbreeding due to the reduced population size could potentially have led to purging of maladapted alleles. This would have increased the fitness of the population emerging from the bottleneck and thus perhaps provided a base for faster positive selection from the freshwater alleles at adaptive loci (i.e. quicker allele frequency changes). Beyond this, we are not aware of any other arguments that could be made regarding the link between homozygosity and haploblock expansion/allele frequency increases.

6. As recombination could theoretically break apart large freshwater haploblocks over successive generations, it would be useful if the authors could briefly discuss whether any evidence of haploblock erosion (e.g., internal recombination events, partial block loss) was observed by SC2017 or SC2020.

Recombination may indeed (almost certainly) have broken some freshwater haploblocks apart. However this signal would have been overwhelmed by the rapid increase in average per-individual freshwater content resulting from increased rate of mating between individuals with

existing freshwater haploblocks, which was still continuing in SC2020 as seen by the increased homozygous freshwater alleles at this time point. As far as we are aware, the only way we would possibly be able to observe internal recombination events or block loss in our data is if we had parent-offspring trios to directly these processes. Unfortunately, despite a lot of pairwise relatedness in our sample, we do not have any trios to fully address the reviewer's observation.

7. While the authors clearly state that Scout Lake was cleared of predators and competitors through rotenone treatment before stickleback introduction, they do not discuss how this simplified ecological context might have influenced the rapid adaptation process observed. The absence of ecological interactions such as predation and interspecific competition could fundamentally alter selective pressures, potentially making Scout Lake a special case. I recommend that the authors explicitly address this point in the Discussion and temper the generalization of their findings accordingly.

We completely agree with the Reviewer that Scout Lake may not capture the full ecological context of a typical case of freshwater adaptation, given that it was cleared of certain competitors prior to the founding. It is something we [the authors] discussed a lot during the preparation of the manuscript, and how to best present our specific results. As a consequence throughout the original manuscript, we attempted to guard against over-generalization of our findings. In the Abstract, for instance, we used the phrase “*an instance of freshwater adaptation...*” [line 65] to denote that our results pertain to Scout Lake. At the beginning of the Discussion, we stated, “*We studied the rapid adaptation of an anadromous Threespine Stickleback population to a new freshwater environment...*”. And in the conclusion we noted, “*Studies of other genomic time series from other Threespine Stickleback will determine how general our results are...*” Thus we believe we have been careful with generalizing our findings, despite realizing the importance of these findings to our understanding of rapid adaptation in threespine stickleback, as well as its implications to evolutionary biology in general.

However, we have now also added the following sentence [lines 532-535] to state this limitation of our study more explicitly.

“While our experimental set up mimicked natural colonizations as much as possible, we note that Scout Lake represented a somewhat simplified ecological context as it was originally cleared of certain competitors through rotenone treatment prior to the founding of the Threespine Stickleback population, even though some of these were re-stocked. Future work utilizing alternative study designs will be essential to determine how generalizable our results are likely to be.”

Reviewer #2 (Remarks to the Author):

This study by Kwakye and colleagues investigates the genomic mechanisms of rapid freshwater adaptation in the Threespine Stickleback. Using time-series genomic data, they propose that adaptation was primarily driven by rare "jackpot" individuals present in the founders, who carried large, pre-existing haploblocks of freshwater-adaptive alleles. These individuals and their descendants are suggested to have rapidly increased in frequency following a population bottleneck, leading to inbreeding and potential purging of deleterious alleles. This proposed mechanism offers a valuable contrast to the traditional transporter hypothesis.

Major Suggestions:

1- The central claim regarding the indispensability of jackpot carriers for the observed rapid sweep would be substantially strengthened by quantitative modelling. Employing explicit ABC or forward simulations could test this hypothesis against alternatives and help disentangle selection from genetic drift, particularly during the bottleneck.

We have now included a forward simulation testing two scenarios with and without jackpot founders, which we believe have significantly strengthened our manuscript, and thank the reviewer for their suggestion. We performed Wright-Fisher simulations in SLiM mimicking the transporter hypothesis, as in Galloway et al. 2020⁴ and Roberts Kingman et al. 2021¹. We let the set up run for 1000 generations, after which we founded a new freshwater environment, representing the Scout Lake population. In one scenario, we found the new lake with both jackpot carriers and non-jackpot individuals, and in another removed the jackpot carriers. Our results showed that the rapid adaptation that we observed empirically in Scout Lake could only be replicated in the presence of jackpot carriers. Even one jackpot carrier was enough for rapid adaptation. We have added a new section in the main paper to describe these simulations and the results [lines 447-473] and also added the simulation setups and details to the supplementary information [supplementary section 15]

2- the argument for early physiological adaptation preceding morphological changes, currently based largely on negative evidence, would be more compelling with direct positive evidence. Measuring relevant physiological traits, such as osmoregulation or metabolic rates, in early-generation individuals could provide this. Or tune down it or move to the supplementary to save text in the main text (see below).

We agree that the results of this section are not that compelling (though it provides some important motivation for future studies beyond the scope of this paper). We have now moved the section on morphological phenotypes and the corresponding discussion to the supplementary texts accordingly, which also helps address the comment below regarding article length.

3- the manuscript's length and density could be reduced for clarity and impact. Now it is too long for the journal. Streamlining the narrative to focus on the core "jackpot carrier" hypothesis and condensing or moving less central results, such as the detailed morphology and fads2 findings, to supplementary information is recommended.

As recommended, we have reduced the length and density of the manuscript. The original submission had 8005 words (abstract plus main text—introduction, results, discussions and conclusion). We have removed and/or moved certain sections to the supplementary. We have removed the sections on “Rapid freshwater adaptation was not sex-specific” and moved “Phenotypes of jackpot carriers and non-jackpot individuals in SC2014” to the supplementary. We have also removed some of our discussions on “The consequences of bottlenecks on genetic load during rapid adaptation”.

In addition to these major changes, we have also moved/removed certain paragraphs as follows:

Moved to supplementary:

“An alternative scenario for the presence of recent hybrids between anadromous and freshwater stickleback in the Scout Lake samples is that the jackpot carriers were produced by matings between anadromous founders and very rare freshwater individuals from the original Scout Lake population that had survived the rotenone treatment of the lake. We tested this hypothesis by creating hybrids from established freshwater populations from the Pacific Coast and Rabbit Slough individuals (see Supplementary Section 13). The distribution of freshwater-adaptive alleles in these hybrids was 50 to 63% compared to the 12.5% to 49.5% content we found in the individuals we sampled in the SC2014 sample, suggesting that, the jackpot carriers observed in SC2014 sample were probably recent oceanic-freshwater hybrids, or their progeny. These were likely to have been present among the Rabbit Slough founders and not hybrids between survivors from the original freshwater Scout Lake population and released anadromous individuals. This latter scenario is further supported by previous studies that have found jackpot carriers in samples collected from marine environments.”

Removed:

“First-degree relatives can either be parents-offspring pairs or siblings. Second-degree relatives include grandparent-grandchild pairs and avuncular pairs. Third-degree relatives include first cousins, great aunts/uncles, grandniece/nephews, and great-grandparents. We can distinguish the two types of first-degree relatives but not the various types of second or third-degree relatives”

“The individual with notable contiguous blocks of freshwater-adaptive alleles in the SC2013 sample was a third-degree relative of an individual in the SC2014 sample with a similar proportion of freshwater alleles. No relatives from this pair were found in the SC2015 sample or later, and this lineage presumably died out in 2014 or failed to proliferate enough to be represented in subsequent samples, possibly because they possessed too few freshwater-adaptive alleles. Thus, their freshwater content of ~7% was insufficient to contribute to the growing population from 2014 and beyond, justifying our criteria of defining a jackpot carrier as individuals with $\geq 10\%$ freshwater-adaptive alleles.”

“Our kinship analysis demonstrated that jackpot carriers mated with non-jackpot individuals early on in the adaptive process, prior to the bottleneck in 2014. In order to examine the extent to which founding non-jackpot individuals contributed freshwater-adaptive alleles to the later post-bottleneck population, we identified loci where all 21 jackpot carriers found in the SC2014 sample were homozygous for the oceanic allele. There were four such unlinked loci. These loci included an 11Kb locus on chromosome IV, a 2Kb locus on chrIX, and 11Kb and 19Kb loci on chrXIX. We then determined if any individual in the subsequent timepoints (SC2015, SC2017 and SC2020) had inherited freshwater alleles at these loci. We found only five heterozygous individuals across all four loci in all the subsequent samples (N=212). However, there were no freshwater adaptive alleles in the non-jackpot individuals observed in the SC2013 and SC2014 samples at these four loci. This indicates that non-jackpot individuals within the founders contributed minimally to the pool of adaptive alleles circulating after the bottleneck.”

These adjustments together with other editing have resulted in reduction of the manuscript length to **5977** words in the main text (which now meets the word limit for the journal) compared to the original 8005.

Minor Suggestions:

1- To improve accessibility for a broader audience, the authors should enhance the in-text explanation of Fig. 6, more explicitly guiding readers through the interpretation of SFS patterns and their connection to conclusions about genetic load and selection.

We have added the following sentences to lines **[413-417]** to guide readers on the SFS interpretations we make:

“In a SFS, the proportion of rare alleles can reflect the demographic history of a population, with excess rare alleles suggesting population growth, while a deficit may suggest population decline. This pattern results from each individual introducing unique mutations to the population as it grows, and vice versa during decline. Demographic processes are also likely to affect both 0- and 4-fold sites equally, while natural selection is more likely to impact 0-fold sites.”

2- all custom scripts and bioinformatic pipelines used for the genomic analyses should be made available.

All custom scripts have been deposited to the first author's GitHub and can be accessed through the following link: <https://github.com/a-kwaky/Rare-Jackpot-Individuals-Drive-Rapid-Adaptation-in-Threespine-Stickleback>. We have also deposited these scripts and intermediate datasets that can be used to replicate the figures as a capsule on Code Ocean and can be accessed once the manuscript is published <https://codeocean.com/capsule/2139373/tree>

3- The figure legend for Fig. 1F: the 344 freshwater-adaptive loci, should briefly explained.

We have added a short description of the 344 freshwater-adaptive loci to the legend of figure 1.

4- The Y-axis labels for the two columns in Fig. 2 need to be clarified.

We have updated the Y-axis label to the sample names instead of year since founding to be consistent with the text and other figures.

5- Fig 4 panel labels should be annotated

This has been corrected.

References

1. Roberts Kingman, G. A. *et al.* Predicting future from past: The genomic basis of recurrent and rapid stickleback evolution. *Sci Adv* **7**, (2021).
2. Kurz, M. L., Heins, D. C., Bell, M. A. & von Hippel, F. A. Shifts in life-history traits of two introduced populations of threespine stickleback. *Evol. Ecol. Res.* **17**, 225–242 (2016).
3. Hendry, A. P. *et al.* Designing eco-evolutionary experiments for restoration projects: Opportunities and constraints revealed during stickleback introductions. *Ecol. Evol.* **14**, e11503 (2024).
4. Galloway, J., Cresko, W. A. & Ralph, P. A Few Stickleback Suffice for the Transport of Alleles to New Lakes. *G3* **10**, 505–514 (2020).

RESPONSE TO ADDITIONAL REVIEWER COMMENTS FOR:

RARE JACKPOT INDIVIDUALS DRIVE RAPID ADAPTATION IN THREESPINE STICKLEBACK” (NCOMMS-25-20702B)

We want to thank the reviewer for their additional comments, which we believe have tremendously improved our manuscript. Our responses to the reviewer’s comments can be found below and any text added to the manuscript are marked **red**.

1. The authors argue that the RS2019 sample was not used to identify adaptive loci and that allele frequencies are highly correlated between RS2009 and RS2019. However, this response does not fully address the original concern. The key issue is not the origin of adaptive loci per se, but whether the RS2019 sample can reliably serve as a reference for standing variation prior to the 2011 introduction event. The authors demonstrate that adaptive alleles underwent selection between 2011 and 2015, raising the possibility that some allele frequency changes also occurred in the RS population over the same period. Given this context, using RS2019 to represent ancestral allele frequencies without any temporal correction or bootstrapping introduces potential biases, even if correlations appear high. More importantly, the RS2009 sample is explicitly available (as acknowledged), and its allele frequencies were used in previous publications. There is no compelling reason not to replicate key frequency-based visualizations (e.g., Fig. 1F) using RS2009 data, or at least provide results based on both datasets to confirm robustness. Without this, the choice of RS2019 remains methodologically suboptimal and weakens the interpretation of standing genetic variation.

As suggested by the reviewer, we have included the RS2009 sample to Fig. 1F (Review Figure 1), Fig. 2 (Review Figure 2) and Fig.5 (Review Figure 3). We have also added the RS2009 sample to the estimated site frequency spectra (Review Figure 4) as supplementary figure S7. We additionally estimated the relatedness between the individuals in the RS2009 sample and the level of inbreeding. As observed in RS2019, we found no related individuals and no evidence of inbreeding in the RS2009 sample. The number of freshwater adaptive alleles in RS2009 ranged from 0.4% to 2% while RS2019 ranged 0% to 1.3% (Review Figure 5). In addition, all the individuals from both RS2009 and RS2019 are assigned to the non-jackpot category by our new BGMM (see below). Overall, both RS2009 and the RS2019 samples appear to be a good reference for standing variation in the ancestral population, with the RS2019 having an advantage of having a much larger sample size for downstream analysis such as assessing the degree of relatedness and quantifying the very low frequency of jackpot individuals present in the marine population.

We now include RS2009 in all our data interpretation regarding the ancestral marine population and relevant figures and have also added the paragraph below at the beginning of the results to indicate we use two Rabbit Slough samples spanning the sampling period in Scout Lake.

Lines 110- 113 *“In addition, we included two samples from the anadromous source population, Rabbit Slough, collected in 2009 and 2019. These two timepoints from the Rabbit Slough represent the genomic diversity in the ancestral population over the same period as examined in Scout Lake. We denote these two Rabbit Slough samples as RS2009 and RS2019. Despite being collected ten years apart, the two Rabbit Slough samples showed highly correlated allele frequencies (Fig. S2 and S3) and highly consistent results when used as baselines for inferring processes in the ancestral oceanic population.”*

Review Figure 1

Review Figure 2

Review Figure 3

Review Figure 4

Review Figure 5

2. The classification of “jackpot carriers” using a fixed 10% threshold for freshwater-adaptive alleles is central to multiple downstream analyses, yet the rationale for this cutoff remains insufficiently justified. While the authors briefly explored sensitivity to alternative thresholds (8% and 12%), these appear arbitrarily selected and do not constitute a systematic evaluation. Moreover, Figure 2C shows that the freshwater content distribution shifts substantially across years, suggesting that a fixed threshold may capture biologically distinct subsets of individuals at different timepoints. This raises concerns about the comparability and interpretation of “jackpot” status across temporal scales. The authors are encouraged to provide a stronger conceptual and statistical justification for the 10% threshold, and to consider whether a relative or quantile-based definition (e.g., top x%) might yield more robust or interpretable results given the temporal dynamics of freshwater allele accumulation.

The 10% cut off was based on the bimodal distribution of the samples collected in 2014, which suggested to us that two distinct groups were in our sample (Review Figure 6). To make this

clearer, we have updated the visualization of the distribution of freshwater adaptive alleles in each year, which was originally presented as a violin plot in supplementary figure S2 (Review Figure 6) with the scatter plot in Review Figure 7, as this shows the distinct groups sampled in SC2014 more clearly. We include Review Figure 7 in a multi-panel Extended Figure (see below).

However, we recognize the reviewer's strong point that our original approach was a subjective grouping and that may introduce biases that affect our downstream interpretation (although we note that the definition of jackpot versus non-jackpot is simply an interpretive tool and is not actually implemented in any downstream analytical frameworks, other than the analysis comparing the AFS of jackpot and non-jackpot individuals for the 2014 time point). Therefore, to make this inference more robust we have distinguished these two potential groups using a more explicit model-based approach.

First we performed a dip test to check the modality of the SC2014 sample based on freshwater content (i.e. number of freshwater alleles per individual) and identified two modes (Review Figure 8).

Secondly, we pooled all samples across timepoints for the same metric of the number of freshwater alleles per individual, and used a Bayesian Gaussian Mixture Modelling (BGMM) approach implemented in scikit-learn (version 1.7.1) in python 3.11.13 to estimate the number of underlying distributions that best fit the data. BGMM incorporates a Dirichlet process prior that allows the model to infer the effective number of components directly from the data. We set the maximum number of components to four and used the full covariance type, which treats each component to have its own general covariance matrix. However, doubling the number of maximum components did not change the result of the analysis. Bayesian information criterion (BIC) suggested two components to be the best choice, consistent with the dip test (Review Figure 9). The fitted model provided posterior probabilities for each observed individual's membership in each of the two main components based on their freshwater content.

We then categorized all the datapoints at each timepoint into one of these two components (Review Figure 10). We obtained a remarkably similar profile of individuals assigned to the two components at each time point compared to that estimated using our original 10% cut off (Review Figure 11), with a Kolmogorov–Smirnov test suggesting that the two distributions were not statistically different (K-S test: statistic = 0.143, p-value = 1). Thus the two components appear to reflect the two categories that we previously called non-jackpot carriers and jackpot carriers using the 10% threshold (Review Figure 11). The model categorized individuals with 5% freshwater adaptive alleles as jackpot carriers with a posterior probability of 0.95. Individuals with 8% freshwater adaptive alleles were classified as jackpot carriers with 100% certainty (Review Figure 12).

We have now adjusted all our results and analysis using this new BGMM-based definition of jackpot and non-jackpot individuals, though as it results in assignments very similar to the original ones, it does not change any of our downstream interpretation. In particular, we have updated Fig. 4 and Fig. S17 to reflect the assignment changes. The major inference from Fig. 4 that involved jackpot and non-jackpot classification was the suggestion that jackpot carriers mated with non-jackpot individuals early on in the colonization process. After applying the BGMM classification, this inference is still valid as we observed relatedness between the two groups. The SFS of jackpot vs non-jackpot carriers in Fig. S17 did not change either. We also repeated our Wright-Fisher simulations using a 5% cut off based on the BGMM model rather than the original 10% cutoff, with our initial result that jackpots are necessary for rapid freshwater adaptation still being robust.

Note that we have now removed all reference to using 10% cutoff in the manuscript as well as our previous testing of alternate thresholds and rely completely on the assignments from the BGMM which have a more robust statistical justification, as requested by the reviewer. Overall we thank the reviewer for pushing us to address this issue and believe our inference is on firmer ground because of it.

In addition to these changes, we have included/modified the following paragraphs to the results and methods to indicate these additional analyses:

Extended Figure 1 showing the distribution of freshwater adaptive alleles across the years and the BGMM predictions for each individual.

Results:

“Therefore, to aid our understanding of genomic change during adaptation to freshwater, we categorized individuals as jackpot carriers and non-jackpot individuals using Bayesian Gaussian Mixture Modeling (BGMM) (Methods, Supplementary Section 9, Extended Fig. 1B). The BGMM categorized individuals with 5% freshwater adaptive alleles as jackpot carriers with a posterior probability of 0.95 and those with 8% freshwater adaptive alleles were classified as jackpot carriers with 100% certainty (Fig. S4). There were no jackpot carriers in RS2009 and RS2019 samples and one jackpot carrier in SC2013 sample (i.e., 1%). There were 23 out of 48 in SC2014 (i.e., 48%), 84 out of 96 in SC2015 (i.e., 88%), and all in SC2017 and SC2020 (i.e., 100%, Table SI). We observed that 27 out of the 47 individuals in SC2014 had contiguous blocks with freshwater-adaptive alleles. These contiguous blocks were mostly heterozygous (Fig. S6) and spanned 0.07Mb [0.044cM] to 22.33Mb [70.48cM], with a mean of 2.40Mb [2.9cM] (Fig. 2A, Supplementary table 2).”

Materials and methods:

“Classification of individuals as jackpot carriers: We performed a dip test to check the modality of the SC2014 sample based on freshwater content (i.e. number of freshwater alleles per individual) and identified two modes. We pooled all samples across timepoints for the same metric of the number of freshwater alleles per individual, and used a Bayesian Gaussian Mixture Modelling (BGMM) approach implemented in scikit-learn (version 1.7.1) in python 3.11.13 to estimate the number of underlying distributions that best fit the data. BGMM incorporates a Dirichlet process prior that allows the model to infer the effective number of components directly from the data. We set the maximum number of components to four and used the full covariance type, which treats each component to have its own general covariance matrix. However, doubling the number of maximum components did not change the result of the analysis. Bayesian information criterion (BIC) suggested two components to be the best choice, consistent with the dip test (Fig. S7). The fitted model provided posterior probabilities for each observed individual’s membership in each of the two main components based on their freshwater content. We then categorized all the datapoints at each timepoint into one of these two components (Fig. S7). The model categorized individuals with 5% freshwater adaptive alleles as jackpot carriers with a posterior probability of 0.95. Individuals with 8% freshwater adaptive alleles were classified as jackpot carriers with 100% certainty.”

Review Figure 6

Review Figure 7

Review Figure 8

Review Figure 9

Review Figure 10

Review Figure 11

Review Figure 12

Extended Figure 1

Revised Figure 4

Revised Fig. S17

Revised Figure 6

3. The authors acknowledge that elevated homozygosity resulting from inbreeding may have facilitated the purging of maladaptive alleles, thus potentially accelerating positive selection at freshwater-adaptive loci. However, this response does not fully address the potential mechanistic implications of increased homozygosity on haploblock structure and allele frequency dynamics. For instance, greater homozygosity could reduce effective recombination, leading to the apparent expansion or preservation of extended haplotypes over time. Additionally, in small populations, hitchhiking effects may be more pronounced, allowing non-adaptive alleles to rise in frequency due to linkage with adaptive loci. These possibilities warrant at least brief acknowledgment in the discussion to avoid oversimplification of the observed genomic patterns as purely selection-driven. A more nuanced consideration of demographic and genetic drift effects would strengthen the interpretation of haploblock dynamics over time.

We believe that we are a little bit clearer about the point the reviewer is making here and appreciate the need for increased nuance in our text. However we note that there is an important distinction between the increased rate of inbreeding and the increase in freshwater haploblock frequency. The former is indeed likely resulting in increased homozygosity across the entire genome as measured by increased F values. Recessive deleterious recessive variants may be purged as a result of this process genomewide, as has been seen in other species going through such extreme bottlenecks with inbreeding. However the freshwater-adaptive haploblocks are actually increasing such that the adaptive loci are transitioning from being primarily homozygous for the marine haplotype to being heterozygous for both freshwater and marine haplotypes, as represented by the long stretches of yellow in Figure 1. Only much later do such loci start to become blue, representing homozygosity of freshwater haploblocks, and even then they are less common than heterozygous haploblocks. Thus effective recombination rates (i.e. observable

recombination events) would not be reduced during this stage, and the rise in freshwater haploblocks must therefore be attributed to differential increases in fitness of freshwater haploblocks (i.e. positive selection). It is true however that while recessive deleterious alleles in most of the genome may be purged due to inbreeding, deleterious alleles in and around freshwater haplotypes may hitchhike, counterbalancing to some extent the purging of genetic load elsewhere in the genome. We have edited/added the following paragraphs to the discussion to reflect these complex dynamics.

“Previous studies have proposed that population bottlenecks may be important during freshwater adaptation by oceanic Threespine Stickleback^{1,2}. These bottlenecks have been inferred from an observed reduction in genetic diversity in established freshwater populations¹⁻³. Here, we present the first direct observation of the temporal dynamics of stickleback adaptation to freshwater during the first nine years of colonization. We observed a bottleneck that occurred during the third and fourth year after founding, likely as a result of reduced fitness of non-jackpot individuals that made up the majority of the founding population. The resulting reduction in population size coincided with a major shift in the genetic composition of the individuals that remained in the lake, indicating the population decline was non-random and driven by strong positive selection for a small number of jackpot carriers.

However, population growth from a few related individuals will decrease genetic variation and potentially increase the population’s genetic load. Genetic load can either be expressed phenotypically or masked⁴ in diploid populations, while bottlenecks can increase phenotypic expression of a population’s genetic load⁴⁻⁶ by exposing rare, recessive deleterious alleles normally masked in heterozygotes. In addition, inbreeding after the bottleneck could increase the frequency of rare deleterious alleles and their probability of homozygosity^{4,5,7}. Such a scenario can lead to one of two outcomes: a) the population could be at risk of extinction due to the increase in the fixation probability of recessive deleterious mutations⁸⁻¹⁰ through drift, or b) negative selection could be more effective at eliminating slightly deleterious mutations, as selection is more potent against recessive or partially recessive mutants in the homozygous state¹¹.

In Scout Lake, the latter outcome could, therefore, increase the population’s overall adaptability by removing deleterious variants that circulated in the anadromous ancestor alongside the positive selection of freshwater-adaptive alleles, at least during the early stages of the adaptive process, before new deleterious mutations emerge. Such a process has been observed in many other species undergoing extreme bottlenecks¹²⁻¹⁴. Although inbreeding may have led to a general purging of deleterious mutations genome-wide, it is interesting to contrast its effects at adaptive loci with regards to the dynamics of freshwater haploblocks. While much of the genome is becoming increasingly homozygous during these earliest stages of freshwater adaptation, adaptive loci are becoming more genetically diverse as homozygous oceanic haploblocks are

*transitioning into heterozygous haploblocks as the freshwater alleles are increasingly positively selected for. This may allow deleterious alleles found within or closely linked to freshwater haplotypes to hitchhike to higher frequencies, thus counteracting the decrease in load elsewhere in the genome. The low recombination rates keeping such haploblocks intact over long periods may act somewhat like inversions, which indeed have been shown to accumulate genetic load in other species such as *Heliconus*¹⁵. This may even slow the speed of fixation of freshwater loci later in the adaptive process as these deleterious alleles become homozygous, again analogous to frequency-dependent selection observed in *Heliconus*, though unlike inversions, the presence of recombination hotspots between adaptive loci may allow some rescue from high genetic load. Indeed, even in established populations we rarely see individuals who are fully homozygous for all freshwater alleles and there are now emerging empirical studies that explore the dynamics of deleterious alleles linked to adaptive loci in low recombination regions (such as in threespine sticklebacks)^{16,17}.”*

4. While the authors reasonably argue that increasing homozygosity could obscure recombination signals, the potential erosion of large freshwater haploblocks remains an important consideration. Even in the absence of parent-offspring trios, population-level patterns, such as reduced haploblock lengths, internal heterogeneity, or linkage disequilibrium decay, could provide indirect but informative evidence of recombination. Currently, the manuscript presents no empirical analysis to support the conclusion that recombination played a limited role. Given that recombination is a central process influencing haplotype integrity and adaptive allele assembly, a lack of data-driven support weakens the strength of this claim. A brief analysis or at least an expanded discussion of potential haploblock erosion would strengthen the interpretation that haploblock persistence reflects biological processes (e.g., selection, demographic dynamics) rather than methodological limitations or overlooked recombination events.

We have now performed statistical tests to determine the differences in overall haploblock lengths from one timepoint to another. We first used Welch's t-test to determine whether the mean haploblock lengths are significantly different from one timepoint to another. There was no significant difference in the mean haploblock lengths between the two Rabbit Slough samples (RS2009 vs RS2019; $t = 1.2021$, $p\text{-value} = 2.3111e-01$), as well as between the RS2019 and SC2013 ($t = 0.4335$, $p\text{-value} = 6.6493e-01$). However, there was a statistically significant difference between the mean haploblock lengths of SC2013 and SC2014 ($t = -9.7188$, $p\text{-value} = 3.4958e-21$), SC2014 and SC2015 samples ($t = -2.2363$, $p\text{-value} = 2.5478e-02$) and SC2015 and SC2017 ($t = -2.7869$, $p\text{-value} = 5.3432e-03$). There was no statistically significant difference between the mean haploblock lengths of SC2017 and SC2020 ($t = -1.4509$, $p\text{-value} = 1.4724e-01$). We further explored the differences in haploblock lengths using Cohen's D, which expresses the mean difference relative to the pooled standard deviation. We estimated Cohen's D and

bootstrapped confidence interval for each pair of consecutive timepoints. There was a small difference between RS2009 and RS2019 ($D=-0.1585[-0.352 - 0.087]$) and RS2019 and SC2013 ($D=0.0044[-0.1341 - 0.26]$) compared to SC2013 and SC2014 ($D=0.4025[0.367 - 0.4510]$). Between SC2014 and SC2015, there was a small difference in the mean haploblock lengths ($D=0.093[0.012 - 0.172]$) compared to SC2013 and SC2014, suggesting that despite the statistically significant t-test, the difference was not substantial at this time period. We observed similar Cohen's D between haploblock means in subsequent years: SC2015 and SC2017 ($D=0.083[0.024 - 0.138]$); SC2017 and SC2020 ($D=0.074[-0.025 - 0.174]$). These results suggest that there was limited increase in the haploblock lengths across the time periods as mediated by recombination, especially after the bottleneck in 2014.

As suggested by the reviewer we have also estimated linkage disequilibrium (LD) decay in and across the freshwater adaptive loci. As previously described for other analyses, we focus on the 280 loci without any missing data. For each time point, we estimated pairwise r^2 among all combinations of the 280 loci using hard-called locus-based genotypes, coded as 0 for homozygous freshwater, 1 for heterozygous, and 2 for homozygous marine, corresponding to the number of marine alleles carried by each individual at each locus. To quantify the uncertainty in LD estimates, we performed bootstrapping using genotype matrices for each of the timepoints. For each timepoint, the genotype matrix was resampled with replacement 1,000 times to generate a distribution of r^2 values in order to estimate 95% confidence intervals (Review Figure 13, 14 and 15). We also estimated the r^2 between individual SNPs (rather than using the genotype call from the whole adaptive region) found in the 344 adaptive regions based on the imputed genotypes, and plotted the mean and standard error (Review Figure 16, 17 and 18). We then binned pairs of loci/SNPs by distance to examine differences in the rate of LD decay.

As RS2009 and SC2020 are much higher coverage than the other population samples (while also containing less samples), they will have more accurate genotype calls compared to the imputed low coverage samples, particularly at true heterozygote sites, leading to systematic difference in r^2 that reflect genotype call accuracy. This effect is visible when comparing the LD decay curves based on individual SNPs for RS2009 vs RS2019, which shows a slight elevation for the former (Review Figure 18). Therefore for the individual SNP-based LD analysis RS2009 and SC2020 were analyzed separately. However, the loci-based estimates of LD will be more robust to such effects as they utilize our approximate likelihood calling approach that leverages information from multiple SNPs at each loci, and indeed RS2009 and RS2019 largely overlap, and thus both low and high coverage genomes are analyzed together in this case.

We have also attempted to test for evidence of partial block loss (haploblock erosion) in our data using an approach that tried to leverage the expected number of meioses separating biologically related individuals in our dataset. Unfortunately, due to the low coverage nature of our data, it

was not possible to distinguish true haplotype disintegration from individual genotype uncertainty.

We have updated the text with the following sentences to reflect these observations:
We also include Review Figure 15 as an extended figure.

Results:

“There was no significant difference in the mean haploblock lengths between the two Rabbit Slough samples (RS2009 vs RS2019; $t = 1.2021$, $p\text{-value} = 2.3111e-01$), as well as between the RS2019 and SC2013 ($t = 0.4335$, $p\text{-value} = 6.6493e-01$). There was a statistically significant difference between the mean haploblock lengths of SC2013 and SC2014 ($t = -9.7188$, $p\text{-value} = 3.4958e-21$), SC2014 and SC2015 samples ($t = -2.2363$, $p\text{-value} = 2.5478e-02$) and SC2015 and SC2017 ($t = -2.7869$, $p\text{-value} = 5.3432e-03$). There was no statistically significant difference between the mean haploblock lengths of SC2017 and SC2020 ($t = -1.4509$, $p\text{-value} = 1.4724e-01$). We further explored the differences in haploblock lengths using Cohen’s D, which expresses the mean difference relative to the pooled standard deviation. We estimated Cohen’s D and bootstrapped confidence interval for each pair of consecutive timepoints. There was a small difference between RS2009 and RS2019 ($D = -0.1585[-0.352 - 0.087]$) and RS2019 and SC2013 ($D = 0.0044[-0.1341 - 0.26]$) compared to SC2013 and SC2014 ($D = 0.4025[0.367 - 0.4510]$). Between SC2014 and SC2015, there was a small difference in the mean haploblock lengths ($D = 0.093[0.012 - 0.172]$), suggesting that despite the statistically significant t-test, the difference was not substantial at this time period. We observed similar Cohen’s D between mean haploblock means between SC2015 and SC2017 ($D = 0.083[0.024 - 0.138]$) as well as SC2017 and SC2020 ($D = 0.074[-0.025 - 0.174]$). These results suggest that there was limited increase in the haploblock lengths across the time periods, especially after the bottleneck in 2014, which would be expected if the haploblocks were being reassembled by recombination.

Venu et al. ¹⁸ indicated that as many as half of all chromosomes in male, as well as a third of chromosomes in females, could be inherited without any recombination event per generation. In addition, recombination is suppressed in individuals that are heterozygous for marine and freshwater haplotypes at adaptive loci. Our sampling period covered nine years, spanning 5-6 generations, and the majority of the individuals sampled after the bottleneck were predominantly heterozygous at freshwater adaptive loci. We estimated linkage disequilibrium (LD) across freshwater adaptive loci (Methods) to assess whether recombination had any substantial effect on the integrity of haploblocks. We observed an elevated levels of LD from SC2014 decay compared to SC2013 and samples from the ancestral population, consistent with a strong directional positive selection increasing the frequency of adaptive haploblocks or potentially the effect of demographic bottleneck that occurred in 2014 (Extended Fig. 2). In subsequent years after the bottleneck, we found no observable patterns of reduction in the rate of LD decay,

suggesting that there was not enough time for recombination to have substantial effect on the integrity of haploblocks.”

Methods

“Linkage disequilibrium (LD) decay: To estimate the LD decay across the sampling period, we focused on the 280 adaptive loci without any missing data. For each time point, we estimated pairwise r^2 among all combinations of the 280 loci using hard-called locus-based genotypes, coded as 0 for homozygous freshwater, 1 for heterozygous, and 2 for homozygous marine, corresponding to the number of marine alleles carried by each individual at each locus. To quantify the uncertainty in LD estimates, we performed bootstrapping using genotype matrices for each of the timepoints. For each timepoint, the genotype matrix was resampled with replacement 1,000 times to generate a distribution of r^2 values in order to estimate 95% confidence intervals. We then binned pairs of loci/SNPs by distance to examine differences in the rate of LD decay.”

Review Figure 13

Review Figure 14

Review Figure 15

LD decay across years using 33,000 SNPs in adaptive regions

Review Figure 16

LD decay across years using 33,000 SNPs in adaptive regions

Review Figure 17

LD decay across years using 33,000 SNPs in adaptive regions

Review Figure 18

References

1. Bell, M. Evolution of phenotypic diversity in *Gasterosteus aculeatus* superspecies on the Pacific coast of North America. *Systematic Biology* **25**, 211–227 (1976).
2. Withler, R. E. & McPhail, J. D. Genetic variability in freshwater and ananorous sticklebacks (*Gasterosteus aculeatus*) of southern British Columbia. *Can J. Zool* **63**, 528–533 (1985).
3. Taylor, E. B. & McPhail, J. D. Historical contingency and ecological determinism interact to prime speciation in sticklebacks, *Gasterosteus*. *Proc. Biol. Sci.* **267**, 2375–2384 (2000).
4. Bertorelle, G. *et al.* Genetic load: genomic estimates and applications in non-model animals. *Nat. Rev. Genet.* **23**, 492–503 (2022).
5. Crow, J. F. Genetic Loads and the Cost of Natural Selection. in *Mathematical Topics in Population Genetics* (ed. Kojima, K.-I.) 128–177 (Springer Berlin Heidelberg, Berlin, Heidelberg, 1970).
6. Wang, J., Hill, W. G., Charlesworth, D. & Charlesworth, B. Dynamics of inbreeding depression due to deleterious mutations in small populations: mutation parameters and inbreeding rate. *Genet. Res.* **74**, 165–178 (1999).
7. Nietlisbach, P., Muff, S., Reid, J. M., Whitlock, M. C. & Keller, L. F. Nonequivalent lethal equivalents: Models and inbreeding metrics for unbiased estimation of inbreeding load. *Evol. Appl.* **12**, 266–279 (2019).
8. Hedrick, P. W. Purging inbreeding depression and the probability of extinction: full-sib mating. *Heredity* **73** (Pt 4), 363–372 (1994).
9. Lynch, M., Conery, J. & Burger, R. Mutation Accumulation and the Extinction of Small Populations. *Am. Nat.* **146**, 489–518 (1995).

10. Palkopoulou, E. *et al.* Complete genomes reveal signatures of demographic and genetic declines in the woolly mammoth. *Curr. Biol.* **25**, 1395–1400 (2015).
11. Barrett, S. C. H. & Charlesworth, D. Effects of a change in the level of inbreeding on the genetic load. *Nature* **352**, 522–524 (1991).
12. Grossen, C., Guillaume, F., Keller, L. F. & Croll, D. Purging of highly deleterious mutations through severe bottlenecks in Alpine ibex. *Nat. Commun.* **11**, 1001 (2020).
13. Croft, L., Matheson, P., Butterworth, N. J. & McGaughran, A. Fitness consequences of population bottlenecks in an invasive blowfly. *Mol. Ecol.* e17492 (2024).
14. Dussex, N., Morales, H. E., Grossen, C., Dalén, L. & van Oosterhout, C. Purging and accumulation of genetic load in conservation. *Trends Ecol. Evol.* **38**, 961–969 (2023).
15. Jay, P. *et al.* Publisher Correction: Mutation load at a mimicry supergene sheds new light on the evolution of inversion polymorphisms. *Nat. Genet.* **53**, 763 (2021).
16. Harringmeyer, O. S. & Hoekstra, H. E. Chromosomal inversion polymorphisms shape the genomic landscape of deer mice. *Nat. Ecol. Evol.* **6**, 1965–1979 (2022).
17. Nickel, J., Laine, J. & Foote, A. D. Complex patterns of hitchhiking mutation load among stickleback populations. *bioRxiv* (2025) doi:10.1101/2025.07.24.666323.
18. Venu, V. *et al.* Fine-scale contemporary recombination variation and its fitness consequences in adaptively diverging stickleback fish. *Nat. Ecol. Evol.* **8**, 1337–1352 (2024).